# Examining the mediating effects of metabolic syndrome components on the relationship between dairy product consumption and nonalcoholic fatty liver disease in Korean adults

**Hyunyu Jeon**[1], **Soo-Hyun Kim**[2], **Dayeon Shin**[2]*

**1** Department of Food and Nutrition, Inha University, Incheon, Republic of Korea, **2** Department of Nutritional Science and Food Management, Ewha Womans University, Seoul, Republic of Korea

* dayeonshin@ewha.ac.kr

## Abstract

### Objective

Dairy products are known to improve blood lipid profiles and insulin sensitivity and to reduce risk factors for metabolic syndrome (MetS) and nonalcoholic fatty liver disease (NAFLD). However, the mechanisms through which dairy product consumption influences NAFLD via MetS components remain unclear. This study examined the mediating effects of MetS components on the association between dairy product consumption and NAFLD.

### Methods

This study included 12,775 Korean adults from the Korea National Health and Nutrition Examination Survey (KNHANES) 2019–2021. Dairy product intake was assessed using a 24-hour dietary recall. NAFLD was defined using a hepatic steatosis index score >36, and MetS was classified according to the National Cholesterol Education Program Adult Treatment Panel III criteria. Multivariable logistic regression analyses were conducted to examine the associations among dairy intake, NAFLD, and MetS components. Mediation analyses with bootstrapping (n = 1,000) were performed to investigate the mediating effects of individual MetS components on the association between dairy consumption and NAFLD.

### Results

Consumption of more than one serving of dairy products was associated with a lower prevalence of NAFLD among women (adjusted odds ratio [AOR], 0.75; 95% confidence interval [CI], 0.59–0.96). Regarding MetS components, intake of one serving of dairy products was associated with lower odds of elevated triglycerides in men (AOR, 0.75; 95% CI, 0.63–0.89). In women, consumption of at least one serving was

**Data availability statement:** This study used publicly available datasets from the 2019-2021 Korea National Health and Nutrition Examination Survey (KNHANES), which can be accessed through the KNHANES official website (https://knhanes.kdca.go.kr/knhanes/main.do). Others would be able to access these data in the same manner as the authors, and the authors did not have any special access privileges that others would not have.

**Funding:** This work was supported by the National Research Foundation of Korea (NRF) grant funded by the Korea government (MSIT) (RS-2024-00340086). There was no additional external funding received for this study. The funders had no role in study design, data collection and analysis, decision to publish, or preparation of the manuscript.

**Competing interests:** The authors have declared that no competing interests exist.

associated with decreased hyperglycemia (AOR, 0.84; 95% CI, 0.73–0.97), abdominal obesity (AOR, 0.69; 95% CI: 0.55–0.87), low high-density lipoprotein cholesterol (AOR, 0.83; 95% CI: 0.72–0.95), and elevated triglyceride levels (AOR, 0.71; 95% CI, 0.60–0.85). Mediation analyses indicated that, among women, significant proportions of the associations between dairy product consumption and NAFLD were mediated by waist circumference (58.0%), systolic blood pressure (18.2%), and high-density lipoprotein cholesterol (51.7%), whereas no significant mediation effects were observed among men.

## Conclusions

Dairy product consumption was associated with a lower prevalence of MetS and NAFLD among women. Mediation analysis further suggested that dairy product consumption may reduce the risk of NAFLD by improving metabolic dysfunction among women.

## Introduction

Nonalcoholic fatty liver disease (NAFLD) is a global health concern, and its worldwide prevalence increased from 25.5% in 2005 to 37.8% in 2016 [1]. In Korea, the prevalence of NAFLD was estimated to reach 30.3% by 2021 [2]. Metabolic syndrome (MetS) is a major contributor to NAFLD, and excess free fatty acids (FFAs) [3], oxidative stress [3], and insulin resistance [4] are well-established risk factors for NAFLD. Insulin resistance promotes the direct influx of glucose into the de novo lipogenesis (DNL) pathway, leading to increased triglyceride synthesis. Consequently, elevated FFAs derived from DNL contribute to hepatic fat accumulation, which plays a central role in the development of NAFLD [5,6].

Dietary factors contribute to the development of NAFLD, including the consumption of dairy products [7,8]. One study reported that individuals in the highest dairy product intake group had a 0.86-fold lower risk of developing NAFLD than those in the lowest intake group [9]. In contrast, a meta-analysis examining food groups and NAFLD reported no significant association between dairy product consumption and NAFLD [10]. Similarly, no association was identified between dairy-related dietary diversity scores and NAFLD [11]. Although dairy product consumption may reduce the risk of NAFLD, existing results remain inconsistent. This inconsistency may be partly because the serving size of dairy products is not standardized internationally, dairy consumption varies by country, and different types of dairy products are consumed.

Previous studies have shown that the nutritional components of dairy products, including whey protein and calcium, may reduce the risk of MetS development. Whey protein supplementation at doses of 0, 5, 10, and 20 g has been shown to decrease postprandial glycemia in a dose-dependent manner in healthy participants [12]. Additionally, dairy proteins act as precursors of angiotensin-I-converting enzyme inhibitory peptides, which have the potential to lower blood pressure [13]. Calcium in dairy products may also confer metabolic benefits by binding bile acids and increasing

cholesterol conversion to bile acids when intestinal reabsorption is inhibited [13]. Therefore, both the protein and calcium content in dairy products demonstrate beneficial effects on the components of MetS.

Previous studies have demonstrated that effective management of MetS may reduce the NAFLD risk [14–17]. Dairy products containing whey protein and calcium have been associated with a decreased risk of MetS through various mechanisms, including lowering insulin resistance, controlling blood glucose levels, reducing blood pressure, and improving the lipid profile. However, the relationship between dairy intake and NAFLD remains inconclusive, with several studies showing no inverse association [9–11,18,19]. Furthermore, the specific mechanisms through which dairy consumption affects MetS components and subsequently influences NAFLD risk remain unclear. Therefore, the aim of this study was to investigate the association between dairy product consumption and NAFLD prevalence and to examine whether NAFLD risk can be reduced through dairy consumption and MetS management. Specifically, the objectives of this study were to 1) determine the association between dairy product intake and NAFLD, 2) examine the effects of dairy product consumption on MetS components, 3) evaluate the effects of MetS components on NAFLD, and 4) examine the mediating effect of MetS components on the relationship between dairy product intake and NAFLD among Korean adults.

## Materials and methods

### Dataset and study participants

This study used data from the Korea National Health and Nutrition Examination Survey (KNHANES) conducted between 2019 and 2021. Among the 22,559 respondents who participated in the KNHANES during this period, individuals aged <19 years (n = 3,868) and those who were pregnant or lactating (n = 100) were excluded. Participants with missing information on covariates, including education level, employment status, household income, marital status, alcohol consumption, smoking status, physical activity, and body mass index (BMI) (n = 2,330), were excluded. Additionally, participants with a total energy intake (<500 kcal or >5,000 kcal; n = 2,982) were excluded. Furthermore, participants who lacked data on MetS components, for whom the hepatic steatosis index (HSI) could not be calculated (n = 307), and those with a history of hepatitis (n = 197) were further excluded. The final sample comprised 12,775 participants (Fig 1). Written informed consent was obtained from all participants before study participation. The dataset was obtained from the Korea Disease Control and Prevention Agency, and the study protocol was approved by the Institutional Review Board (IRB approval no. 2018-01-03-C-A, 2018-01-03-2C-A, and 2018-01-03-5C-A).

### Assessment of dairy product intake

Daily consumption of dairy products was computed from a dietary intake survey using a 1-day 24-hour dietary recall. Dairy products were identified using tertiary food codes and categorized as condensed milk, milk, goat milk, liquid yogurt, semi-solid yogurt, sorbet, ice milk, ice cream, cheese, or cream. These items were converted into serving sizes according to the dietary reference intakes of Koreans [20]. One serving size was defined as 200 g of milk (including condensed milk, milk, and goat milk), 150 g of liquid yogurt, 100 g of semi-solid yogurt, ice cream (including sorbet, ice milk, and ice cream), cream, and 20 g of cheese. Dairy product intake was classified as "0" for no intake, "0–1" for intake of less than one serving, and "≥1" for intake of one or more servings.

### Assessment of nonalcoholic fatty liver disease

The hepatic steatosis index (HSI) is a simple and widely used screening tool for NAFLD. The HSI formula used in this study was as follows: HSI = 8 × (ALT/AST ratio) + BMI + 2 (women) + 2 (diabetes mellitus) [21]. Alanine aminotransferase (ALT) and aspartate aminotransferase (AST) levels were measured using a LaboSpect 008AS (Hitachi, Tokyo, Japan) after fasting for at least 8 h. BMI was calculated as weight (kg) divided by height ($m^2$). Height was measured using a Seca 274 stadiometer (Seca, Hamburg, Germany), and weight using a GL-6000–20 scale (G-tech, Seoul, Republic of Korea).

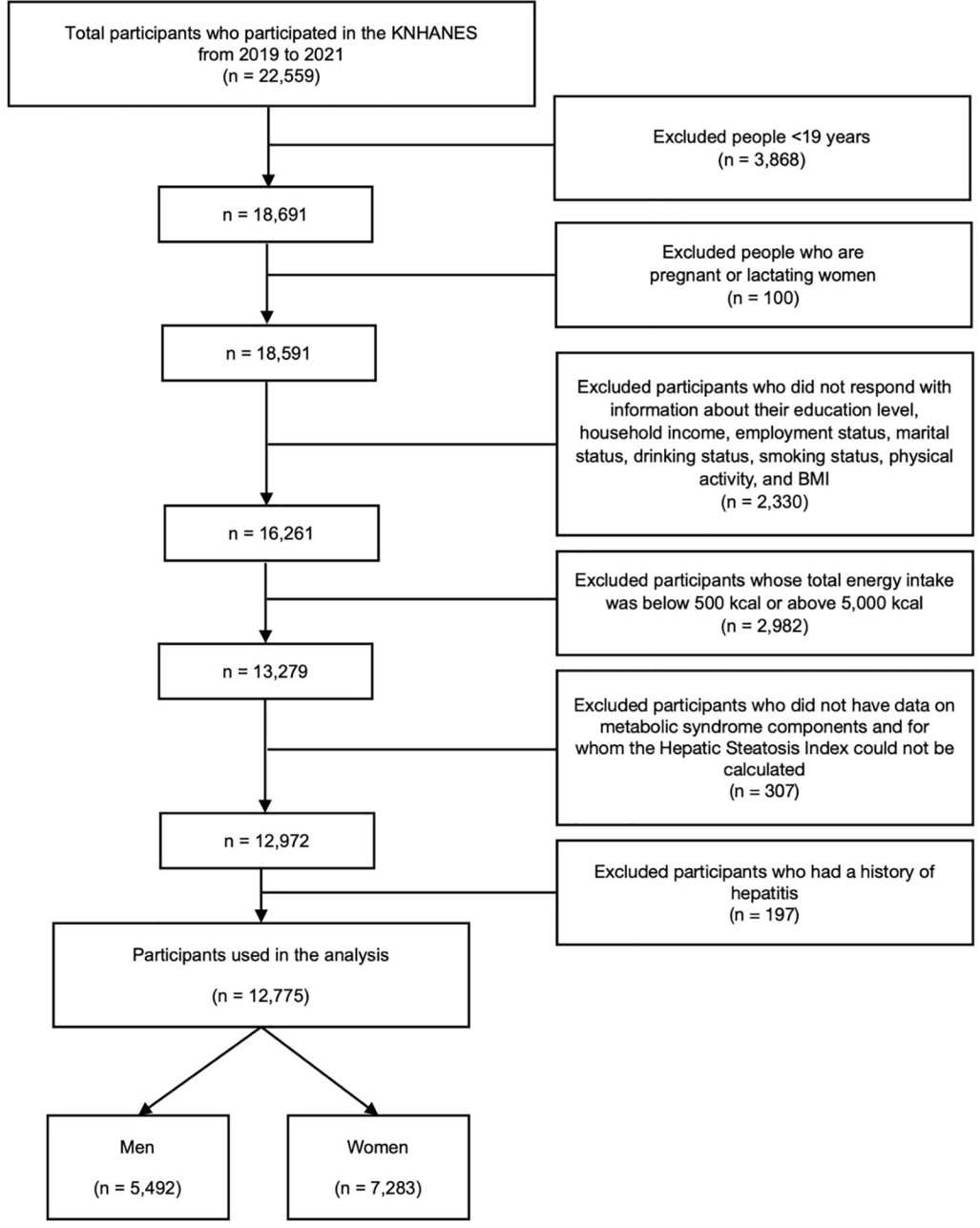

**Fig 1. Flow diagram of study participant selection.**

Data regarding sex and diabetes status were collected using a self-administered survey. NAFLD was defined using the HSI with a cutoff value of >36.

## Assessment of metabolic syndrome components

According to the National Cholesterol Education Program Adult Treatment Panel III (NCEP ATP III) guidelines, MetS was defined as present when an individual met at least three of the following five criteria: (1) fasting blood glucose ≥100 mg/

dL or current use of medication or insulin treatment for hyperglycemia or type 2 diabetes; (2) waist circumference ≥90 cm for men and ≥80 cm for women; (3) systolic blood pressure (SBP) ≥130 mmHg or diastolic blood pressure (DBP) ≥85 mmHg, or current use of antihypertensive medication; (4) HDL cholesterol <40 mg/dL for men and <50 mg/dL for women; and (5) triglycerides ≥150 mg/dL [22]. Blood samples were collected from participants after a fasting period of at least 8 h. HDL cholesterol, triglyceride, and fasting blood glucose levels were measured using a Labospect 008AS (Hitachi, Tokyo, Japan). Waist circumference was assessed at the midpoint between the lowest rib and highest iliac crest using a Lufkin W606 tape measure (Lufkin, USA). SBP and DBP were measured using a non-mercury aneroid sphygmomanometer (WatchBP Office AFIB; Microlife, Switzerland), and the average of these measurements was used.

## Assessment of covariates

Demographic characteristics included sex, age, BMI, education level, household income, employment status, marital status, household type, and region. Sex was classified as "men" or "women," and education level was categorized as "≤elementary school," "middle school," "high school," or "≥college." Household income was categorized into quartiles as "low" (quartile 1), "low-medium" (quartile 2), "medium-high" (quartile 3), and "high" (quartile 4). Employment status was classified as "employed" or "unemployed," and marital status was classified as "single" or "married." Household type was categorized as "single-person" or "multi-person," and region was classified as "urban" or "rural." Health behaviors included alcohol consumption, smoking status, and physical activity. Alcohol consumption was classified as "none" (never or <1 time/month), "moderate" (1–4 times/month), or "high" (more than 2 times/week). Smoking status was classified as "never," "past," or "current." Physical activity was classified as "yes" or "no" based on engagement in at least 2.5 h of moderate-intensity activity, 1.25 h of vigorous-intensity activity, or a combination of both per week.

## Statistical analyses

The complex sampling design of the survey was determined by applying sampling weights, strata, and clusters. For analyses combining multiple survey years, the integrated weight was calculated by multiplying the sampling weight for each year by the integration ratio. To compare general characteristics according to NAFLD status, continuous variables are presented as means and standard errors (SEs) using t-tests and general linear models, and categorical variables are presented as frequencies (n) and percentages (%) using chi-square tests. Multivariable logistic regression was performed to estimate the adjusted odds ratios (AORs) and 95% confidence intervals (CIs) for the association of dairy product intake with NAFLD, dairy product intake with MetS components, and MetS components with NAFLD. To analyze the association between MetS components, each MetS component was categorized into two groups using the same criteria applied in the definition of MetS. All models were adjusted for age, education level, household income, employment status, marital status, household type, region, alcohol consumption, smoking status, total energy intake, physical activity, and BMI. To evaluate the mediating effects of MetS components on the relationship between dairy product consumption and NAFLD, multiple regression analyses were conducted using a bootstrapping method in an SAS macro (n = 1,000). All variables included in the mediation analysis were treated as continuous. MetS components were analyzed as continuous clinical measurements, and NAFLD was represented by a continuous HSI score. Using a continuous HSI score allowed us to capture the full spectrum of hepatic steatosis severity, preserve variability, and satisfy the linearity assumptions of the bootstrapping approach implemented in the SAS macro. All statistical analyses were conducted using SAS software (version 9.4; SAS Institute, Cary, NC, USA). Two-sided $p$-values < 0.05 were considered statistically significant.

## Results

Table 1 presents the general characteristics of the study participants according to the NAFLD prevalence. NAFLD was observed in 29.17% of men and 19.27% of women. The mean age was higher in the NAFLD group than in the control group for both men and women (all $p < 0.001$). BMI was higher in the NAFLD group compared with the normal group in

**Table 1. General characteristics of the study participants with the presence of non-alcoholic fatty liver disease (NAFLD).**

| Characteristics | Men (n = 5,492) | | p-value | Women (n = 7,283) | | p-value |
|---|---|---|---|---|---|---|
| | NAFLD | Normal | | NAFLD | Normal | |
| **Participants (n, %)** | 1,450 (29.17%) | 4,042 (70.83%) | | 1,525 (19.27%) | 5,758 (80.73%) | |
| **Age (years)** | 48.72±0.38 | 43.03±0.40 | <0.001 | 52.05±0.51 | 47.58±0.31 | <0.001 |
| **BMI (kg/m²)** | 28.21±0.10 | 23.35±0.05 | <0.001 | 28.67±0.13 | 22.09±0.05 | <0.001 |
| **Education level (n, %)** | | | <0.001 | | | <0.001 |
| ≤Elementary school | 108 (4.37%) | 612 (9.55%) | | 462 (21.79%) | 1,172 (14.09%) | |
| Middle school | 90 (4.55%) | 439 (8.00%) | | 189 (11.03%) | 534 (7.42%) | |
| High school | 544 (38.37%) | 1,423 (37.98%) | | 498 (38.08%) | 1,790 (33.66%) | |
| ≥College | 708 (52.71%) | 1,568 (44.47%) | | 376 (29.09%) | 2,262 (44.83%) | |
| **Household income (n, %)** | | | <0.01 | | | <0.001 |
| Low | 179 (9.92%) | 777 (13.98%) | | 355 (17.92%) | 1,063 (13.84%) | |
| Low-medium | 325 (21.46%) | 990 (22.40%) | | 429 (27.10%) | 1,361 (22.38%) | |
| Medium-high | 436 (30.79%) | 1,064 (28.67%) | | 396 (28.43%) | 1,571 (29.49%) | |
| High | 510 (37.84%) | 1,211 (34.95%) | | 345 (26.55%) | 1,763 (34.29%) | |
| **Employment status (n, %)** | | | <0.001 | | | 0.05 |
| Employed | 1,105 (77.73%) | 2,728 (71.54%) | | 775 (51.58%) | 3,064 (54.73%) | |
| Unemployed | 345 (22.27%) | 1,314 (28.46%) | | 750 (48.42%) | 2,694 (45.27%) | |
| **Marital status (n, %)** | | | <0.001 | | | <0.001 |
| Single | 405 (34.14%) | 852 (28.66%) | | 143 (13.81%) | 931 (22.55%) | |
| Married | 1,045 (65.86%) | 3,190 (71.34%) | | 1,382 (86.19%) | 4,827 (77.45%) | |
| **Household type (n, %)** | | | 0.76 | | | 0.83 |
| Single-person households | 194 (12.62%) | 564 (12.25%) | | 234 (11.02%) | 845 (10.83%) | |
| Multi-person households | 1,256 (87.38%) | 3,478 (87.75%) | | 1,291 (88.98%) | 4,913 (89.17%) | |
| **Region (n, %)** | | | 0.14 | | | <0.0001 |
| Urban | 1,166 (86.39%) | 3,174 (84.58%) | | 1,155 (82.44%) | 4,688 (87.13%) | |
| Rural | 284 (13.61%) | 868 (15.42%) | | 370 (17.56%) | 1,070 (12.87%) | |
| **Alcohol consumption (n, %)** | | | <0.001 | | | <0.001 |
| None | 500 (33.66%) | 1,349 (31.46%) | | 1,026 (64.95%) | 3,436 (55.04%) | |
| Moderate | 565 (40.29%) | 1,354 (36.16%) | | 377 (26.48%) | 1,699 (33.19%) | |
| High | 385 (26.06%) | 1,339 (32.38%) | | 122 (8.57%) | 623 (11.77%) | |
| **Smoking status (n, %)** | | | 0.04 | | | 0.28 |
| Never | 381 (27.80%) | 1,031 (27.71%) | | 1,329 (85.70%) | 5,117 (87.57%) | |
| Past | 597 (38.36%) | 1,845 (42.13%) | | 106 (7.54%) | 374 (6.92%) | |
| Current | 472 (33.84%) | 1,166 (30.15%) | | 90 (6.76%) | 267 (5.50%) | |
| **Physical activity (n, %)** | | | 0.09 | | | <0.01 |
| Yes | 799 (53.63%) | 2,171 (50.76%) | | 979 (61.76%) | 3,390 (56.35%) | |
| No | 651 (46.37%) | 1,871 (49.24%) | | 546 (38.24%) | 2,368 (43.65%) | |
| **Dairy products (g/day)** | 79.89±4.41 | 76.20±2.85 | 0.47 | 79.01±4.86 | 100.03±2.68 | <0.001 |
| **Types of dairy products (g/day)** | | | | | | |
| Milk | 54.80±3.93 | 52.05±2.36 | 0.56 | 54.98±4.37 | 69.45±2.32 | <0.01 |
| Liquid yogurt | 3.34±0.82 | 3.10±0.41 | 0.79 | 3.04±0.55 | 3.23±0.30 | 0.77 |
| Semisolid yogurt | 10.93±1.57 | 11.15±0.92 | 0.90 | 12.44±1.37 | 16.53±0.93 | 0.01 |
| Ice cream | 8.36±1.00 | 7.61±0.66 | 0.52 | 6.68±1.16 | 7.64±0.54 | 0.43 |
| Cheese | 1.78±0.34 | 1.79±0.19 | 0.97 | 1.02±0.18 | 2.21±0.17 | <0.001 |
| Cream | 0.69±0.19 | 0.51±0.16 | 0.47 | 0.84±0.27 | 0.96±0.14 | 0.70 |

*(Continued)*

**Table 1.** (Continued)

| Characteristics | Men (n = 5,492) | | p-value | Women (n = 7,283) | | p-value |
|---|---|---|---|---|---|---|
| | **NAFLD** | **Normal** | | **NAFLD** | **Normal** | |
| **MetS (n, %)** | | | <0.001 | | | <0.001 |
| Yes | 908 (59.20%) | 1,083 (22.71%) | | 1,062 (66.59%) | 1,295 (17.36%) | |
| No | 542 (40.80%) | 2,959 (77.29%) | | 463 (33.41%) | 4,463 (82.64%) | |
| **Fasting blood glucose (mg/dL)** | 108.33 ± 0.89 | 100.61 ± 0.37 | <0.001 | 110.05 ± 0.87 | 95.13 ± 0.24 | <0.001 |
| **Waist circumference (cm)** | 96.50 ± 0.26 | 84.49 ± 0.15 | <0.001 | 92.81 ± 0.29 | 76.56 ± 0.14 | <0.001 |
| **Blood pressure (mmHg)** | | | | | | |
| SBP | 123.06 ± 0.40 | 119.33 ± 0.29 | <0.001 | 122.97 ± 0.47 | 113.91 ± 0.30 | <0.001 |
| DBP | 80.82 ± 0.31 | 76.20 ± 0.20 | <0.001 | 76.89 ± 0.31 | 72.29 ± 0.18 | <0.001 |
| **HDL cholesterol (mg/dL)** | 44.18 ± 0.30 | 49.92 ± 0.23 | <0.001 | 49.48 ± 0.30 | 57.99 ± 0.22 | <0.001 |
| **Triglycerides (mg/dL)** | 188.75 ± 4.08 | 135.50 ± 2.11 | <0.001 | 143.17 ± 2.46 | 98.43 ± 0.93 | <0.001 |

Categorical variables are presented as numbers (%), and continuous variables are presented as means ± standard errors (SEs).

The p-values were estimated using the t-test for continuous variables and the chi-square test for categorical variables.

NAFLD was defined as an HSI > 36.

Abbreviations: NAFLD, nonalcoholic fatty liver disease; HSI, hepatic steatosis index; BMI, body mass index; MetS, metabolic syndrome; SBP, systolic blood pressure; DBP, diastolic blood pressure; HDL, high-density lipoprotein.

both sexes (all $p < 0.001$). In terms of education level, men were most likely to have college education in both groups, whereas women were most likely to have high school education in the NAFLD group and college education in the normal group (all $p < 0.001$). The most common category of household income was high in both the NAFLD and normal groups among men ($p < 0.01$), whereas among women, household income was medium to high in the NAFLD group and high in the normal group ($p < 0.001$). Employment status showed significant differences according to NAFLD status in both sexes, with a higher proportion of employment in the NAFLD group than in the normal group among men (77.73% vs. 71.54%; $p < 0.0001$), and a higher proportion in the normal group than in the NAFLD group among women (54.73% vs. 51.58%; $p < 0.05$). Alcohol consumption was significantly different according to the presence of NAFLD in both sexes (all $p < 0.0001$), and smoking was significantly different according to the presence of NAFLD only in men ($p = 0.04$). Physical activity was significantly different according to the presence of NAFLD only in women ($p < 0.01$). Consumption of dairy products was not significantly different between the NAFLD group and the normal group in men ($p = 0.47$); meanwhile, for women, dairy product consumption was higher in the normal group (100.03 g/day) than in the NAFLD group (79.01 g/day) ($p < 0.0001$). Consumption of specific dairy products was also not significantly different among men; however, among women, the normal group consumed significantly more milk ($p < 0.01$), semisolid yogurt ($p = 0.01$), and cheese ($p < 0.001$) than the NAFLD group. The prevalence of MetS was significantly higher in the NAFLD group than in the normal group in both sexes (all $p < 0.001$). For clinical measurements of MetS components, the NAFLD group exhibited higher levels of fasting blood glucose, waist circumference, SBP, DBP, and triglycerides and lower levels of HDL cholesterol than the normal group in both men and women (all, $p < 0.001$).

Nutrient intake of the participants according to their NAFLD status is shown in Table 2. The total energy intake was not significantly different between the NAFLD and normal groups in either sex. Regarding the macronutrient energy distribution, the percentage of energy from carbohydrates was higher in the NAFLD group (62.15%) than in the normal group (60.30%) among women ($p < 0.0001$), whereas no significant differences were observed in the percentage of energy from protein across groups in both sexes. Among men, fat intake percentage was higher in the NAFLD group than in the normal group (22.90% vs. 21.92%; $p = 0.002$), whereas among women, fat intake was higher in the normal group than in the NAFLD group (22.96% vs. 21.39%; $p < 0.0001$). Men in the NAFLD group consumed more fat (58.37 g/day; $p = 0.02$) than women consumed in the normal group (42.69 g/day; $p = 0.003$). The intake of saturated, monounsaturated, and

**Table 2. Nutrient intake of the participants by the status of nonalcoholic fatty liver disease (NAFLD) status.**

| Variables | Men (n = 5,492) | | p-value | Women (n = 7,283) | | p-value |
|---|---|---|---|---|---|---|
| | NAFLD | Normal | | NAFLD | Normal | |
| Total energy intake (kcal) | 2,216.14 ± 24.98 | 2,171.38 ± 16.25 | 0.13 | 1,589.03 ± 22.08 | 1,610.99 ± 10.68 | 0.36 |
| % energy of carbohydrate | 56.89 ± 0.43 | 57.67 ± 0.27 | 0.11 | 62.15 ± 0.41 | 60.30 ± 0.21 | <0.0001 |
| % energy of protein | 15.38 ± 0.16 | 15.18 ± 0.09 | 0.25 | 15.03 ± 0.15 | 14.99 ± 0.08 | 0.84 |
| % energy of fat | 22.90 ± 0.28 | 21.92 ± 0.19 | 0.002 | 21.39 ± 0.30 | 22.96 ± 0.17 | <0.0001 |
| Carbohydrate (g/day) | 304.25 ± 3.22 | 301.34 ± 2.14 | 0.44 | 241.66 ± 3.12 | 237.49 ± 1.59 | 0.23 |
| Protein (g/day) | 84.92 ± 1.21 | 82.43 ± 0.80 | 0.08 | 59.74 ± 1.13 | 60.39 ± 0.50 | 0.58 |
| Fat (g/day) | 58.37 ± 1.18 | 55.18 ± 0.75 | 0.02 | 39.50 ± 0.97 | 42.69 ± 0.48 | 0.003 |
| Saturated fatty acid (g/day) | 18.43 ± 0.41 | 17.48 ± 0.26 | 0.04 | 12.54 ± 0.36 | 13.69 ± 0.18 | 0.003 |
| Monounsaturated fatty acid (g/day) | 19.46 ± 0.47 | 18.18 ± 0.30 | 0.02 | 12.69 ± 0.35 | 13.65 ± 0.17 | 0.01 |
| Polyunsaturated fatty acids (g/day) | 14.62 ± 0.31 | 13.89 ± 0.19 | 0.04 | 10.05 ± 0.24 | 10.84 ± 0.13 | 0.003 |
| Dietary fiber (g/day) | 27.38 ± 0.37 | 28.18 ± 0.26 | 0.07 | 23.89 ± 0.40 | 23.24 ± 0.20 | 0.14 |
| Vitamin A (µg RAE/day) | 423.66 ± 14.24 | 414.88 ± 8.65 | 0.60 | 358.73 ± 9.90 | 379.58 ± 5.89 | 0.06 |
| Vitamin E (mg α-TE/day) | 7.90 ± 0.12 | 7.77 ± 0.08 | 0.35 | 6.02 ± 0.10 | 6.28 ± 0.06 | 0.03 |
| Vitamin C (mg/day) | 74.69 ± 6.02 | 65.32 ± 1.69 | 0.12 | 60.55 ± 2.14 | 66.24 ± 1.29 | 0.02 |
| β-Carotene (µg/day) | 2,973.29 ± 77.17 | 3,093.81 ± 72.35 | 0.26 | 2,732.49 ± 85.61 | 2,730.28 ± 46.22 | 0.98 |
| Magnesium (mg/day) | 331.53 ± 4.02 | 339.58 ± 2.65 | 0.10 | 270.33 ± 4.19 | 269.89 ± 1.99 | 0.92 |
| Zinc (mg/day) | 11.76 ± 0.16 | 11.72 ± 0.11 | 0.85 | 8.70 ± 0.14 | 8.71 ± 0.06 | 0.93 |
| Calcium (mg/day) | 532.29 ± 8.50 | 534.26 ± 5.50 | 0.84 | 441.22 ± 8.76 | 460.37 ± 4.36 | 0.04 |
| Phosphorus (mg/day) | 1,196.37 ± 13.75 | 1,198.31 ± 9.55 | 0.91 | 915.38 ± 14.44 | 931.78 ± 6.67 | 0.27 |
| Sodium (mg/day) | 3,958.56 ± 61.80 | 3,803.15 ± 36.49 | 0.03 | 2,772.08 ± .53.79 | 2,726.20 ± 25.46 | 0.44 |
| Potassium (mg/day) | 2,945.66 ± 34.51 | 2,999.71 ± 25.77 | 0.21 | 2,492.41 ± 40.64 | 2,478.52 ± 19.02 | 0.74 |

Values are presented as mean ± standard error (SE).

The p-values were estimated using the t-test for continuous variables.

NAFLD was defined as an HSI > 36.

Abbreviations: NAFLD, nonalcoholic fatty liver disease; HSI, hepatic steatosis index.

polyunsaturated fatty acid total fat was higher in the NAFLD group than in the normal group in men (p = 0.04, p = 0.02, p = 0.04, respectively), whereas it was higher in the normal group than in the NAFLD group in women (p = 0.003, p = 0.01, and p = 0.003, respectively). For micronutrients, dairy vitamin E, vitamin C, and calcium intake were significantly lower in women with NAFLD (p = 0.03, p = 0.02, p = 0.04, respectively), whereas no significant differences were observed among men. Sodium intake was higher in men in the NAFLD group than in those in the normal group (p = 0.03), whereas no significant difference was observed in women.

The association between dairy product intake and NAFLD is presented in Table 3. Women who consumed at least one serving size of dairy products showed a significant inverse association with NAFLD compared with women who never consumed dairy products (AOR, 0.75; 95% CI, 0.59–0.96), whereas no significant association was observed in men. However, when dairy product intake was measured in grams and according to the type of dairy product, no significant association was observed in either sex.

Table 4 shows the association between dairy product intake and MetS components. Among men, consumption of at least one serving of dairy products was associated with significantly lower odds of elevated triglycerides (AOR, 0.75; 95% CI, 0.63–0.89) compared with no dairy consumption. Among women, consumption of 0–1 serving of dairy products was associated with a significantly lower odds of low HDL cholesterol (AOR, 0.79; 95% CI, 0.65–0.96) compared with no dairy

**Table 3. Association between dairy product intake and nonalcoholic fatty liver disease (NAFLD).**

| Men (n = 5,492) | AOR (95% CI) | *p*-value | Women (n = 7,283) | AOR (95% CI) | *p*-value |
|---|---|---|---|---|---|
| **Dairy products (serving size)** | | | **Dairy products (serving size)** | | |
| 0 (n = 3,506) | 1.00 (Ref.) | | 0 (n = 3,953) | 1.00 (Ref.) | |
| 0 to 1 (n = 656) | 1.07 (0.79–1.46) | 0.66 | 0 to 1 (n = 1,113) | 0.88 (0.66–1.17) | 0.39 |
| ≥ 1 (n = 1,330) | 1.02 (0.81–1.28) | 0.89 | ≥ 1 (n = 2,217) | 0.75 (0.59–0.96) | 0.02 |
| **Men (n = 5,492)** | **β ± SE** | **p-value** | **Women (n = 7,283)** | **β ± SE** | **p-value** |
| Dairy products (g/day) | −0.0001 ± 0.0003 | 0.7991 | Dairy products (g/day) | −0.0001 ± 0.0002 | 0.6820 |
| Types of dairy products | | | Types of dairy products | | |
| Milk (g/day) | −0.0001 ± 0.0004 | 0.7722 | Milk (g/day) | −0.0001 ± 0.0002 | 0.6259 |
| Liquid yogurt (g/day) | −0.0008 ± 0.0016 | 0.6095 | Liquid yogurt (g/day) | 0.0004 ± 0.0012 | 0.7584 |
| Semisolid yogurt (g/day) | 0.0000 ± 0.0008 | 0.9570 | Semisolid yogurt (g/day) | 0.0001 ± 0.0005 | 0.8033 |
| Ice cream (g/day) | 0.0003 ± 0.0012 | 0.7827 | Ice cream (g/day) | −0.0004 ± 0.0012 | 0.7417 |
| Cheese (g/day) | 0.0010 ± 0.0071 | 0.8884 | Cheese (g/day) | −0.0007 ± 0.0029 | 0.8096 |
| Cream (g/day) | 0.0003 ± 0.0071 | 0.9667 | Cream (g/day) | 0.0004 ± 0.0035 | 0.9086 |

One serving size of dairy products was defined as 200g for milk, 150g for liquid yogurt, 100g for semisolid yogurt, ice cream, and cream, and 20g for cheese. Dairy product intake was classified as "0" for no intake, "0–1" for consumption exceeding 0g and intake of less than one serving size, and "≥1" for intake of more than one serving size.

Multivariable logistic regression and linear regression were used to estimate the associations between dairy product intake and components of metabolic syndrome, adjusting for age, BMI, education level, occupation, marital status, household income, household type, region, drinking, smoking, total energy intake, and physical activity.

Abbreviations: NAFLD, non-alcoholic fatty liver disease; AOR, adjusted odds ratio; CI, confidence interval; SE, standard error.

consumption. Additionally, women consuming at least one serving of dairy products had significantly lower odds of hyperglycemia (AOR, 0.84; 95% CI, 0.73–0.97), abdominal obesity (AOR, 0.69; 95% CI, 0.55–0.87), low HDL cholesterol (AOR, 0.83; 95% CI, 0.72–0.95), and elevated triglycerides (AOR, 0.71; 95% CI, 0.60–0.85), compared with women who never consumed dairy products.

For every 1g increase in dairy product intake, waist circumference decreased in both men and women (men: β = −0.0009; women: β = −0.001; all *p* < 0.05). Blood pressure also decreased with increasing dairy intake among men, with significantly lower values observed for both SBP (β = −0.003) and DBP (β = −0.002); among women, a decrease was observed for SBP (β = −0.004). Regarding lipid profiles, HDL cholesterol increased by 0.003 mg/dL among women (*p* = 0.01), whereas triglyceride levels decreased by 0.037 mg/dL among men (*p* = 0.0003) and 0.025 mg/dL among women (*p* < 0.0001).

Table 5 presents the AORs and 95% CIs for NAFLD according to MetS components. All components were significantly associated with NAFLD prevalence in both sexes: hyperglycemia (men: 1.99 [1.60–2.48]; women: 2.82 [2.26–3.52]), abdominal obesity (men: 1.90 [1.43–2.52]; women: 1.47 [1.05–2.06]), elevated blood pressure (men: 1.36 [1.11–1.68]; women: 1.68 [1.32–2.14]), low HDL-cholesterol (men: 1.64 [1.32–2.04]; women: 1.97 [1.61–2.41]), and elevated triglycerides (men: 1.97 [1.61–2.42]; women: 2.19 [1.75–2.73]).

Fig 2 illustrates the mediating effects of MetS components on the association between dairy product intake and NAFLD. Among women, significant natural indirect effects (NIEs) of dairy product intake on NAFLD were observed through waist circumference (β = −0.00008; 95% CI −0.00012 to −0.00003), with a proportion mediated of 58.0% (Fig 2B), SBP (β = −0.00002; 95% CI −0.00004 to 0.0000), with a proportion mediated of 18.2% (Fig 2C), and through HDL cholesterol (β = −0.00007; 95% CI −0.00011 to −0.00003), with a proportion mediated of 51.7% (Fig 2E). The NIEs through fasting blood glucose (β = −0.00009; 95% CI −0.00020 to 0.00001) and DBP (β = −0.00002; 95% CI −0.00004 to 0.0000) were not statistically significant, although the corresponding proportions mediated were 72.2% and 14.0%, respectively

**Table 4. Association between dairy product intake and components of metabolic syndrome.**

| Men (n = 5,492) | Hyperglycemia | | Abdominal obesity | | Elevated blood pressure | | Low HDL-cholesterol | | Elevated triglycerides | |
|---|---|---|---|---|---|---|---|---|---|---|
| | AOR (95% CI) | p-value | AOR (95% CI) | p-value | AOR (95% CI) | p-value | AOR (95% CI) | p-value | AOR (95% CI) | p-value |
| Dairy products (serving size) | | | | | | | | | | |
| 0 (n = 3,506) | 1.00 (Ref.) | | 1.00 (Ref.) | | 1.00 (Ref.) | | 1.00 (Ref.) | | 1.00 (Ref.) | |
| 0 to 1 (n = 656) | 1.04 (0.83–1.30) | 0.72 | 0.94 (0.69–1.28) | 0.68 | 0.95 (0.76–1.19) | 0.65 | 1.08 (0.84–1.39) | 0.53 | 0.98 (0.79–1.20) | 0.81 |
| ≥ 1 (n = 1,330) | 0.88 (0.74–1.03) | 0.11 | 0.90 (0.71–1.15) | 0.40 | 0.83 (0.69–1.01) | 0.06 | 0.93 (0.78–1.11) | 0.40 | 0.75 (0.63–0.89) | 0.001 |
| **Women (n = 7,283)** | AOR (95% CI) | p-value | AOR (95% CI) | p-value | AOR (95% CI) | p-value | AOR (95% CI) | p-value | AOR (95% CI) | p-value |
| Dairy products (serving size) | | | | | | | | | | |
| 0 (n = 3,953) | 1.00 (Ref.) | | 1.00 (Ref.) | | 1.00 (Ref.) | | 1.00 (Ref.) | | 1.00 (Ref.) | |
| 0 to 1 (n = 1,113) | 0.83 (0.69–1.01) | 0.06 | 0.86 (0.66–1.13) | 0.27 | 0.92 (0.75–1.14) | 0.43 | 0.79 (0.65–0.96) | 0.02 | 0.84 (0.69–1.01) | 0.07 |
| ≥ 1 (n = 2,217) | 0.84 (0.73–0.97) | 0.02 | 0.69 (0.55–0.87) | 0.002 | 0.96 (0.81–1.13) | 0.60 | 0.83 (0.72–0.95) | 0.01 | 0.71 (0.60–0.85) | 0.0001 |

| Men (n = 5,492) | Fasting blood glucose (mg/dL) | | Waist circumference (cm) | | SBP (mmHg) | | DBP (mmHg) | | HDL cholesterol (mg/dL) | | Triglycerides (mg/dL) | |
|---|---|---|---|---|---|---|---|---|---|---|---|---|
| | β ± SE | p-value | β ± SE | p-value | β ± SE | p-value | β ± SE | p-value | β ± SE | p-value | β ± SE | p-value |
| Dairy products (g/day) | −0.002±0.002 | 0.49 | −0.0009±0.0004 | 0.02 | −0.003±0.001 | 0.02 | −0.002±0.001 | 0.02 | 0.0005±0.001 | 0.68 | −0.037±0.010 | 0.0003 |
| Milk (g/day) | −0.002±0.003 | 0.42 | −0.001±0.0005 | 0.22 | −0.002±0.001 | 0.22 | −0.003±0.001 | 0.02 | 0.001±0.001 | 0.59 | −0.040±0.010 | 0.0001 |
| Liquid yogurt (g/day) | −0.025±0.009 | 0.004 | −0.006±0.002 | 0.01 | −0.020±0.008 | 0.01 | 0.002±0.009 | 0.83 | −0.001±0.006 | 0.83 | 0.092±0.098 | 0.35 |
| Semisolid yogurt (g/day) | 0.0005±0.006 | 0.93 | −0.002±0.001 | 0.05 | −0.003±0.004 | 0.42 | 0.001±0.003 | 0.76 | 0.001±0.003 | 0.73 | −0.057±0.032 | 0.07 |
| Ice cream (g/day) | 0.016±0.014 | 0.23 | −0.001±0.002 | 0.54 | −0.017±0.005 | 0.002 | −0.007±0.004 | 0.11 | −0.004±0.005 | 0.34 | −0.008±0.046 | 0.86 |
| Cheese(g/day) | −0.035±0.018 | 0.05 | −0.006±0.005 | 0.27 | 0.011±0.014 | 0.46 | −0.012±0.015 | 0.43 | −0.001±0.014 | 0.93 | −0.253±0.135 | 0.06 |
| Cream (g/day) | −0.052±0.014 | 0.0001 | −0.002±0.006 | 0.72 | 0.024±0.020 | 0.23 | 0.019±0.014 | 0.16 | 0.034±0.020 | 0.10 | −0.110±0.186 | 0.56 |
| **Women (n = 7,283)** | Fasting blood glucose (mg/dL) | | Waist circumference (cm) | | SBP (mmHg) | | DBP (mmHg) | | HDL cholesterol (mg/dL) | | Triglycerides (mg/dL) | |
| | β ± SE | p-value | β ± SE | p-value | β ± SE | p-value | β ± SE | p-value | β ± SE | p-value | β ± SE | p-value |
| Dairy products (g/day) | −0.002±0.002 | 0.16 | −0.001±0.0004 | 0.03 | −0.004±0.001 | 0.01 | −0.001±0.001 | 0.23 | 0.003±0.001 | 0.01 | −0.025±0.005 | <0.0001 |
| Milk (g/day) | −0.002±0.002 | 0.31 | −0.001±0.0004 | 0.17 | −0.003±0.002 | 0.04 | −0.0005±0.001 | 0.63 | 0.003±0.001 | 0.04 | −0.025±0.006 | <0.0001 |
| Liquid yogurt (g/day) | −0.011±0.009 | 0.21 | −0.002±0.002 | 0.29 | −0.005±0.010 | 0.62 | 0.002±0.007 | 0.84 | 0.005±0.009 | 0.59 | 0.088±0.049 | 0.08 |
| Semisolid yogurt (g/day) | −0.003±0.004 | 0.47 | −0.002±0.001 | 0.02 | −0.005±0.003 | 0.08 | −0.003±0.002 | 0.22 | 0.004±0.003 | 0.22 | −0.038±0.014 | 0.01 |
| Ice cream (g/day) | −0.002±0.005 | 0.72 | −0.0001±0.002 | 0.97 | −0.002±0.005 | 0.70 | −0.009±0.004 | 0.02 | −0.0004±0.005 | 0.93 | −0.015±0.023 | 0.52 |
| Cheese (g/day) | −0.014±0.015 | 0.35 | −0.010±0.005 | 0.07 | −0.007±0.017 | 0.67 | 0.010±0.013 | 0.44 | 0.074±0.019 | 0.0001 | −0.057±0.065 | 0.38 |
| Cream (g/day) | 0.005±0.016 | 0.75 | 0.009±0.006 | 0.12 | −0.018±0.015 | 0.22 | 0.004±0.011 | 0.69 | −0.013±0.023 | 0.57 | −0.018±0.069 | 0.80 |

One serving size of dairy products was defined as 200g for milk, 150g for liquid yogurt, 100 g for semisolid yogurt, ice cream, and cream, and 20g for cheese. Dairy product intake was classified "0 serving size" for 0 intake, "0–1 serving size" for consumption exceeding 0g and less than one serving size, and "≥1 serving size" for intake of more than one serving size.

Multivariable logistic regression and linear regression were used to estimate the associations between dairy product intake and components of metabolic syndrome, adjusting for age, BMI, education level, occupation, marital status, household income, household type, region, drinking, smoking, total energy intake, and physical activity.

Abbreviations: NAFLD, non-alcoholic fatty liver disease; AOR, adjusted odds ratio; CI, confidence interval; HDL, high-density lipoprotein; SE, standard error.

**Table 5. Adjusted odds ratio (AOR) and 95% confidence intervals (CIs) for nonalcoholic fatty liver disease (NAFLD) according to components of metabolic syndrome.**

| Men (n = 5,492) | AOR (95% CI) | *p*-value | Women (n = 7,283) | AOR (95% CI) | *p*-value |
|---|---|---|---|---|---|
| **Hyperglycemia (mg/dL)** | | | | | |
| < 100 (n = 2,908) | 1.00 (Ref.) | | < 100 (n = 4,721) | 1.00 (Ref.) | |
| ≥ 100 (n = 2,584) | 1.99 (1.60–2.48) | <0.0001 | ≥ 100 (n = 2,562) | 2.82 (2.26–3.52) | <0.0001 |
| **Abdominal obesity (cm)** | | | | | |
| < 90 (n = 3,213) | 1.00 (Ref.) | | < 80 (n = 3,576) | 1.00 (Ref.) | |
| ≥ 90 (n = 2,279) | 1.90 (1.43–2.52) | <0.0001 | ≥ 80 (n = 3,707) | 1.47 (1.05–2.06) | 0.02 |
| **Elevated blood pressure (mmHg)** | | | | | |
| SBP < 130 and DBP < 85 (n = 2,920) | 1.00 (Ref.) | | SBP < 130 and DBP < 85 (n = 4,484) | 1.00 (Ref.) | |
| SBP ≥ 130 or DBP ≥ 85 (n = 2,572) | 1.36 (1.11–1.68) | 0.0039 | SBP ≥ 130 or DBP ≥ 85 (n = 2,799) | 1.68 (1.32–2.14) | <0.0001 |
| **Low HDL-cholesterol (mg/dL)** | | | | | |
| > 40 (n = 4,199) | 1.00 (Ref.) | | > 50 (n = 4,783) | 1.00 (Ref.) | |
| ≤ 40 (n = 1,293) | 1.64 (1.32–2.04) | <0.0001 | ≤ 50 (n = 2,500) | 1.97 (1.61–2.41) | <0.0001 |
| **Elevated triglycerides (mg/dL)** | | | | | |
| < 150 (n = 3,555) | 1.00 (Ref.) | | < 150 (n = 5,839) | 1.00 (Ref.) | |
| ≥ 150 (n = 1,937) | 1.97 (1.61–2.42) | <0.0001 | ≥ 150 (n = 1,444) | 2.19 (1.75–2.73) | <0.0001 |

Multivariable logistic regression was used to estimate adjusted odds ratios (AORs) and 95% confidence intervals (CIs), adjusting for age, BMI, education level, occupation, marital status, household income, household type, region, drinking, smoking, total energy intake, and physical activity.

Abbreviations: NAFLD, nonalcoholic fatty liver disease; AOR, adjusted odds ratio; CI, confidence interval; HDL, high-density lipoprotein.

(Figs 2A and 2D). No significant mediation was observed for triglyceride levels (Fig 2F). Among men, all mediated proportions were reported as not applicable (NA) because the estimated values exceeded 100%. Sensitivity analyses using age- and BMI-stratified mediation models were conducted, with a greater number of NIEs observed among women than men across strata (S1 and S2 Tables).

## Discussion

### Summary of study findings

This study examined the mediating effects of MetS components on the association between dairy product consumption and NAFLD in Korean adults. Among women, consumption of at least one serving of dairy products per day was associated with a lower prevalence of NAFLD as well as a reduced prevalence of several MetS components, including hyperglycemia, abdominal obesity, low HDL cholesterol, and elevated triglycerides, compared with women who did not consume dairy products. In contrast, no significant associations were observed among men. Additionally, all MetS components were independently associated with NAFLD in both men and women. Mediation analyses further indicated that the protective association between dairy intake and NAFLD in women was significantly mediated by waist circumference, SBP, and HDL cholesterol levels, whereas no significant mediation effects were observed among men.

### Association between dairy product consumption and NAFLD

The inverse association between dairy product consumption and NAFLD observed in this study is consistent with the findings of previous studies conducted in Asian populations. For example, the consumption of ≥ 50 g of full-fat dairy products was inversely associated with the prevalence of NAFLD in Thailand [18]. Similarly, among Chinese individuals, those who consumed seven servings of dairy products per week exhibited a lower incidence of NAFLD than those who did not consume dairy products [23].

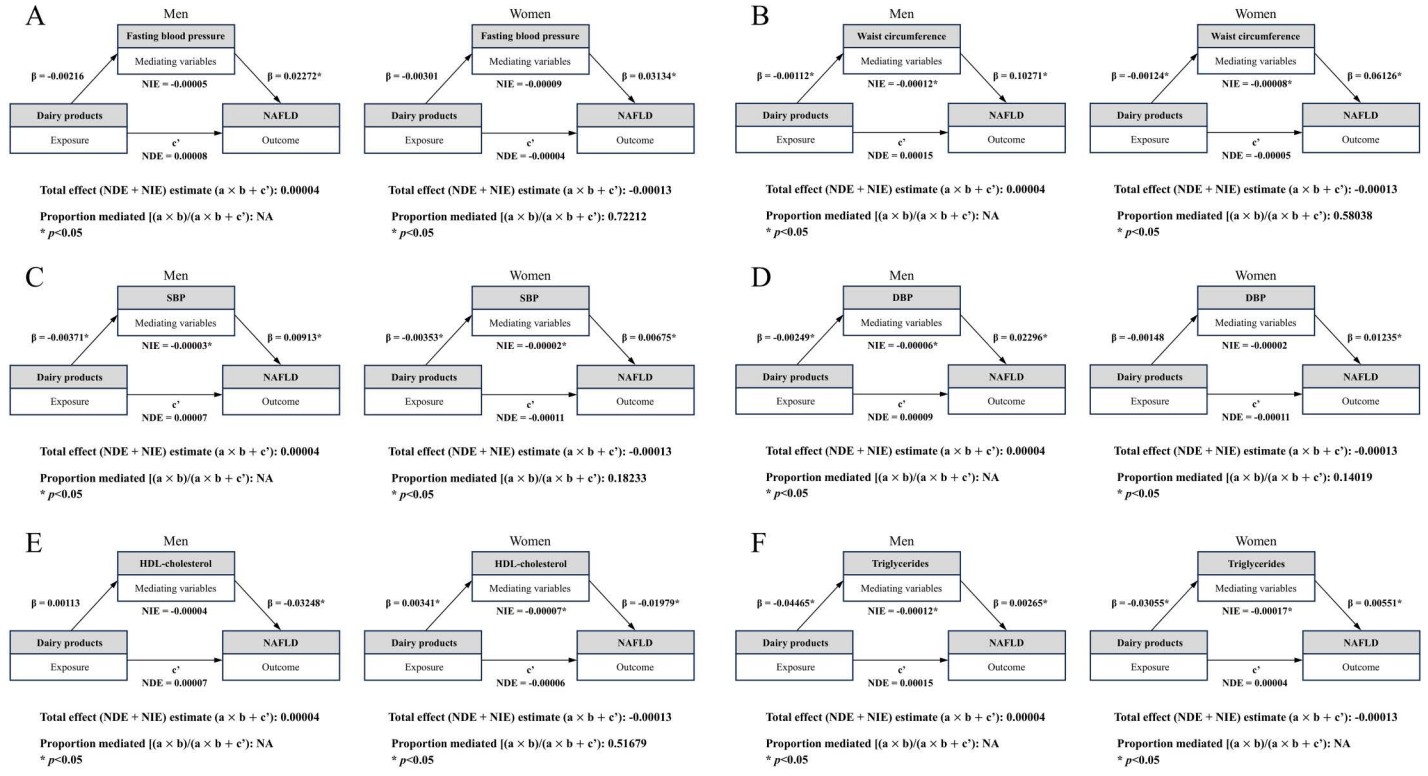

**Fig 2. Model of the mediation relationship between dairy product consumption and nonalcoholic fatty liver disease through metabolic syndrome components as mediating variables.** (A) fasting blood glucose. (B) Waist circumference. (C) SBP. (D) DBP. (E) HDL-cholesterol. (F) Triglycerides. The proportion mediated was calculated as the NIE divided by the total effect and was reported as NA when the value exceeded 1. All models were adjusted for age, BMI, education level, occupation, marital status, household income, household type, region, drinking, smoking, total energy intake, and physical activity. NAFLD, nonalcoholic fatty liver disease; NIE, natural indirect effect; NDE, natural direct effect; SBP, systolic blood pressure; DBP, diastolic blood pressure; HDL, high-density lipoprotein.

Several biological mechanisms can explain this association. Dairy products with high calcium content can reduce the circulating levels of 1,25-dihydroxy vitamin D, resulting in limited uptake of calcium into cells. Lower intracellular calcium levels stimulate lipolysis and inhibit the expression of fatty acid synthase, which in turn inhibits lipid synthesis and prevents fat accumulation in the liver [13]. Additionally, dairy components such as inulin-type fructans and whey protein may modulate the gut microbiota, potentially contributing to the inhibition of NAFLD progression by regulating the gut-liver axis [24].

## Association between dairy product consumption and metabolic syndrome components

The relationship between dairy product consumption and MetS components has also been examined by previous studies [14,18,25,26]. A 200 g/day increase in milk consumption was associated with a 12% lower risk of abdominal obesity, and a 100 g/day increase in yogurt consumption was associated with a 16% lower risk of hyperglycemia [26]. In addition, when dairy product intake was divided into quartiles, individuals in the highest quartile had significantly lower odds of developing abdominal obesity and MetS than those in the lowest quartile [27].

This association can be attributed to several biological pathways. Whey proteins in dairy products increase plasma amino acid levels, including isoleucine, leucine, valine, and lysine [28]. This increase stimulates the production of glucagon-like peptide-1 and gastric inhibitory polypeptide [15], which subsequently enhances insulin secretion and leads to reduced blood glucose levels [16]. Furthermore, dietary calcium intake may increase HDL cholesterol by reducing lipid

absorption via binding to bile acids and fatty acids [17,29]. Moreover, lactotripeptides in dairy proteins inhibit angiotensin-converting enzymes, thereby reducing blood pressure [30].

However, this study did not find a significant association between dairy product consumption and blood pressure. Previous studies have reported inconsistent results regarding this relationship. Nevertheless, several studies have identified an inverse association between dairy intake and blood pressure across diverse age groups, including adults, older adults, children, and adolescents [31–33]. Particularly, low-fat dairy consumption has been reported to be more effective in reducing blood pressure, whereas full-fat dairy products have shown little or no effect [34]. In contrast, a Dutch study reported that overall dairy consumption was not significantly associated with changes in blood pressure [35]. Moreover, a Mendelian randomization analysis incorporating data from 32 observational and cohort studies found no evidence of a causal effect of dairy consumption on SBP or the risk of hypertension [36]. Similarly, no significant association was observed between the consumption of different dairy products and the prevalence or incidence of hypertension [37]. Overall, this existing evidence does not provide a clear consensus on the relationship between dairy consumption and blood pressure. Therefore, further longitudinal and clinical studies are warranted to clarify the potential role of dairy consumption in the regulation of BP.

## Association between metabolic syndrome components and NAFLD

NAFLD and MetS are highly interrelated, with strong associations between NAFLD and individual components of MetS [38]. Insulin resistance disturbs blood glucose control, increases liver fat accumulation, and inhibits lipolysis, leading to increased very-low-density lipoprotein (VLDL) levels and decreased HDL cholesterol levels, thereby contributing to the progression of both MetS and NAFLD [39–41]. Elevated blood pressure is also associated with an increased risk of NAFLD, with individuals with prehypertension and hypertension showing approximately 1.3-fold and 1.8-fold higher odds of NAFLD, respectively, than those with normal blood pressure [42]. Moreover, abdominal obesity has been strongly associated with a higher risk of NAFLD [43], and individuals with NAFLD exhibit significantly higher triglyceride and lower HDL cholesterol levels than those without NAFLD [44].

## Sex-specific associations of dairy consumption with metabolic syndrome and NAFLD

The present study demonstrated that the associations between dairy consumption, NAFLD, and MetS components varied by sex, with significant associations observed only among women. Similar sex-specific patterns have been reported previously. A cohort study found that higher dairy consumption was strongly associated with a lower prevalence of NAFLD in women [45], a finding supported by a meta-analysis of observational studies that reported consistent results [46]. Regarding MetS components, a 10-year cohort study from the Korean Genome and Epidemiology Study (KoGES) showed that dairy consumption was associated with a reduced risk of low HDL-cholesterol among women [47], and a cross-sectional study using the KNHANES observed a lower MetS prevalence among Korean women consuming ≥1 serving/day of dairy products [25]. Moreover, MetS and its individual components are more strongly associated with NAFLD in women than in men [48].

These sex differences may partly arise because the metabolic effects of diets and dietary constituents are expressed in a sex-specific manner and highly dependent on the disease context [49]. Clinical evidence supports the existence of sex-specific metabolic responses to dairy consumption. In a 6-week randomized controlled trial (RCT), low-fat dairy intake differentially improved MetS parameters according to sex, with a modest but significant reduction in fasting blood glucose in men and reductions in waist circumference, body weight, and BMI in women [50]. Another RCT demonstrated distinct postprandial lipoprotein responses to dairy between sexes, with women exhibiting smaller increases in VLDL particles and greater increases in HDL particle concentrations than men [51]. Biological sex also strongly influences adipose tissue distribution and metabolic functions. Women generally have greater amounts of subcutaneous adipose tissue than men [52]. Although abdominal subcutaneous fat is associated with adverse glucose and lipid profiles [53], subcutaneous fat may

provide partial protection against metabolic dysfunction, potentially reducing the risk of MetS in women [54]. In addition, insulin sensitivity differs by sex, with women typically exhibiting greater adipose tissue insulin sensitivity than men [55]. Dairy intake has consistently been shown to inhibit intestinal fat absorption, improve lipid profiles, reduce blood pressure, and enhance glucose metabolism [56–58], thereby contributing to improvements in NAFLD. Sex hormones further modulate these processes, with estrogens promoting metabolically protective pathways and androgens supporting male-specific metabolic environments [59]. Collectively, these biological and clinical differences suggest that dairy exposure may involve distinct metabolic pathways in men and women, contributing to the observed sex-specific associations.

### Sex-specific mediating effects of metabolic syndrome components in the dairy-NAFLD association

The present study found that the association between dairy consumption and NAFLD operates primarily through MetS components, rather than through a direct pathway. Additionally, in women, waist circumference, SBP, and HDL cholesterol appeared to serve as significant mediators of the dairy-NAFLD association, whereas no significant mediating effects were observed in men. This sex-specific mediation pattern supports the idea that the metabolic benefits of dairy intake are expressed differently according to sex. Women in this study consumed more dairy products than men, and consumption of more than one dairy product per day was significantly associated with lower odds of MetS in Korean women [25]. In addition, although dairy consumption did not differ between men in the NAFLD and non-NAFLD groups, women in the non-NAFLD group reported significantly higher dairy consumption than those with NAFLD. This pattern may explain why the beneficial metabolic effects of dairy intake, such as enhanced lipid excretion and lower blood glucose levels, are more pronounced in women. Importantly, the mediation analysis did not identify any significant direct effects of dairy consumption on NAFLD, indicating that dairy intake predominantly influences NAFLD through its effects on several metabolic dysfunctions, including elevated waist circumference, SBP, and low HDL cholesterol levels, rather than through a direct hepatic pathway in women.

### Study strengths and limitations

This study has certain limitations. First, it was a cross-sectional study using the KNHANES dataset, which does not allow for the establishment of a clear causal relationship. Additionally, although the KNHANES dataset includes information on whether participants were taking medication for dyslipidemia, it lacks detailed information regarding medication type, dosage, therapeutic targets (e.g., LDL cholesterol, HDL cholesterol, and triglycerides), and combination therapy. Moreover, the dataset does not provide information on liver fibrosis stage or liver enzyme levels, such as gamma-glutamyl transferase (GGT) or alkaline phosphatase (ALP), which may also influence NAFLD risk. Thus, additional longitudinal studies are needed to clarify the causal relationships among dairy products, NAFLD, and MetS. Furthermore, the use of noninvasive methods for diagnosing NAFLD may result in misclassification and diagnostic errors. Despite these limitations, this study has several strengths. To the best of our knowledge, this is the first study to identify the mediating effects of MetS components on the association between dairy product consumption and NAFLD. The serving sizes used in this study (200 g for milk, 100–150 g for yogurt, and 20 g for cheese) were consistent with the standard portions of international dietary guidelines [26,60,61] and represent realistic targets for clinical dietary counseling. Although dairy product preferences differ across countries, the focus on total dairy intake in the primary mediation analysis supports the broader applicability of our findings. Sensitivity analyses examining individual dairy types allow for interpretation across diverse dietary contexts, although the generalizability of cheese-specific findings may be limited, given the relatively low cheese consumption in our Korean cohort compared to Western populations.

### Conclusions

This study demonstrated that the associations between dairy product consumption and NAFLD and between dairy consumption and individual MetS components were significant among women but not among men. In contrast, the individual MetS components were significantly associated with NAFLD in both sexes. Mediation analyses further indicated that waist

circumference, SBP, and HDL cholesterol levels mediated the association between dairy consumption and NAFLD in women, whereas no significant association was observed in men.

## Supporting information

**S1 Table. Mediation effects of metabolic syndrome components on the association between dairy product consumption and NAFLD, stratified by age.** Statistically significant values are indicated in bold (*$p < 0.05$). $\beta_1$ represents the regression coefficient for the association between dairy product consumption and the metabolic syndrome component. $\beta_2$ represents the regression coefficient for the association between the metabolic syndrome component and NAFLD. The proportion mediated was calculated as NIE divided by the total effect and is reported as NA when the value exceeded 1. All models were adjusted for BMI, education level, occupation, marital status, household income, household type, region, drinking, smoking, total energy intake, and physical activity. NAFLD, non-alcoholic fatty liver disease; NIE, natural indirect effect; NDE, natural direct effect; SBP, systolic blood pressure; DBP, diastolic blood pressure; HDL, high-density lipoprotein. (DOCX)

**S2 Table. Mediation effects of metabolic syndrome components on the association between dairy product consumption and NAFLD, stratified by BMI.** Statistically significant values are indicated in bold (*$p < 0.05$). $\beta_1$ represents the regression coefficient for the association between dairy product consumption and the metabolic syndrome component. $\beta_2$ represents the regression coefficient for the association between the metabolic syndrome component and NAFLD. The proportion mediated was calculated as NIE divided by the total effect and is reported as NA when the value exceeded 1. All models were adjusted for age, education level, occupation, marital status, household income, household type, region, drinking, smoking, total energy intake, and physical activity. NAFLD, non-alcoholic fatty liver disease; NIE, natural indirect effect; NDE, natural direct effect; SBP, systolic blood pressure; DBP, diastolic blood pressure; HDL, high-density lipoprotein. (DOCX)

## Author contributions

**Conceptualization:** Dayeon Shin.

**Data curation:** Hyunyu Jeon, Dayeon Shin.

**Formal analysis:** Hyunyu Jeon, Soo-Hyun Kim, Dayeon Shin.

**Funding acquisition:** Dayeon Shin.

**Investigation:** Dayeon Shin.

**Methodology:** Hyunyu Jeon, Dayeon Shin.

**Project administration:** Dayeon Shin.

**Supervision:** Dayeon Shin.

**Visualization:** Soo-Hyun Kim.

**Writing – original draft:** Hyunyu Jeon, Dayeon Shin.

**Writing – review & editing:** Hyunyu Jeon, Soo-Hyun Kim, Dayeon Shin.

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
