## [Decision Letter · Decision Letter 0]

18 Feb 2025

PONE-D-24-31691Examining mediating effects of metabolic syndrome components on dairy products and non-alcoholic fatty liver diseasePLOS ONE

Dear Dr. Shin,

Thank you for submitting your manuscript to PLOS ONE. After careful consideration, we feel that it has merit but does not fully meet PLOS ONE’s publication criteria as it currently stands. Therefore, we invite you to submit a revised version of the manuscript that addresses the points raised during the review process.

We look forward to receiving your revised manuscript.

Kind regards,

Ian James Martins, PhD

Academic Editor

PLOS ONE

**Journal Requirements:**

1. When submitting your revision, we need you to address these additional requirements.Please ensure that your manuscript meets PLOS ONE's style requirements, including those for file naming. The PLOS ONE style templates can be found at https://journals.plos.org/plosone/s/file?id=wjVg/PLOSOne_formatting_sample_main_body.pdf and https://journals.plos.org/plosone/s/file?id=ba62/PLOSOne_formatting_sample_title_authors_affiliations.pdf 2. Thank you for stating in your Funding Statement: This work was supported by the National Research Foundation of Korea (NRF) grant funded by the Korea government (MSIT) (RS-2024-00340086).Please provide an amended statement that declares *all* the funding or sources of support (whether external or internal to your organization) received during this study, as detailed online in our guide for authors at http://journals.plos.org/plosone/s/submit-now Please also include the statement “There was no additional external funding received for this study.” in your updated Funding Statement. Please include your amended Funding Statement within your cover letter. We will change the online submission form on your behalf.

Reviewers' comments:

Reviewer's Responses to Questions

**Comments to the Author**

1. Is the manuscript technically sound, and do the data support the conclusions?

Reviewer #1: Yes

Reviewer #2: Yes

2. Has the statistical analysis been performed appropriately and rigorously? 

Reviewer #1: Yes

Reviewer #2: Yes

3. Have the authors made all data underlying the findings in their manuscript fully available?

The PLOS Data policy requires authors to make all data underlying the findings described in their manuscript fully available without restriction, with rare exception (please refer to the Data Availability Statement in the manuscript PDF file). The data should be provided as part of the manuscript or its supporting information, or deposited to a public repository. For example, in addition to summary statistics, the data points behind means, medians and variance measures should be available. If there are restrictions on publicly sharing data—e.g. participant privacy or use of data from a third party—those must be specified.requires authors to make all data underlying the findings described in their manuscript fully available without restriction, with rare exception (please refer to the Data Availability Statement in the manuscript PDF file). The data should be provided as part of the manuscript or its supporting information, or deposited to a public repository. For example, in addition to summary statistics, the data points behind means, medians and variance measures should be available. If there are restrictions on publicly sharing data—e.g. participant privacy or use of data from a third party—those must be specified.requires authors to make all data underlying the findings described in their manuscript fully available without restriction, with rare exception (please refer to the Data Availability Statement in the manuscript PDF file). The data should be provided as part of the manuscript or its supporting information, or deposited to a public repository. For example, in addition to summary statistics, the data points behind means, medians and variance measures should be available. If there are restrictions on publicly sharing data—e.g. participant privacy or use of data from a third party—those must be specified.requires authors to make all data underlying the findings described in their manuscript fully available without restriction, with rare exception (please refer to the Data Availability Statement in the manuscript PDF file). The data should be provided as part of the manuscript or its supporting information, or deposited to a public repository. For example, in addition to summary statistics, the data points behind means, medians and variance measures should be available. If there are restrictions on publicly sharing data—e.g. participant privacy or use of data from a third party—those must be specified.

Reviewer #1: Yes

Reviewer #2: Yes

4. Is the manuscript presented in an intelligible fashion and written in standard English?

Reviewer #1: Yes

Reviewer #2: Yes

5. Review Comments to the Author

**Reviewer #1:** Dear Authors,Dear Authors,Dear Authors,Dear Authors,

Your topic” Examining mediating effects of metabolic syndrome components on dairy products and non-alcoholic fatty liver disease” seems to be corrected as follows:

“Effects of dairy products components on metabolic syndrome and non-alcoholic fatty liver disease”

What is Your mean of Mediation analysis in key words?

Please mention mechanisms of dairy effects on metabolic syndrome and NAFLD in the abstract. Also add a brief review of previous study.

It was better you using the 24-h dietary recall third a week (One holiday and two non-holiday days.

Please add statistical methods under all tables.

Add Antioxidants such as vitamin E, ,Zn, Se, Mg, and amount of different kinds of fatty acids and amino acids to table 2.

Please mention others research with more details for comparison their results with Your study results in the discussion.

Regards,

**Reviewer #2:** The research "Examining mediating effects of metabolic syndrome components on dairy products and The research "Examining mediating effects of metabolic syndrome components on dairy products and The research "Examining mediating effects of metabolic syndrome components on dairy products and The research "Examining mediating effects of metabolic syndrome components on dairy products and

non-alcoholic fatty liver disease" is interesting. However, author should discuss their findings rigorously to explain the reason behind the findings. Discussion is weak.

6. PLOS authors have the option to publish the peer review history of their article (what does this mean?). If published, this will include your full peer review and any attached files.). If published, this will include your full peer review and any attached files.). If published, this will include your full peer review and any attached files.). If published, this will include your full peer review and any attached files.

...

Reviewer #1: **Yes:** Professor Naheed AryaeianProfessor Naheed AryaeianProfessor Naheed AryaeianProfessor Naheed Aryaeian

Reviewer #2: No

---

## [Author Response · Author response to Decision Letter 1]

5 Apr 2025

Response letter to Reviewer 1

1) What is Your mean of Mediation analysis in key words?

Response: The authors thank the reviewer’s comment.

Mediation analysis examines the underlying mechanism of the observed relationship between an exposure variable and an outcome variable, exploring how these two variables are related through a third intermediate variable called the mediator. Rather than assuming a direct causal relationship between the independent and dependent variables, mediation analysis hypothesizes that the exposure variable influences the mediator, which in turn affects the outcome variable. Through this process, the mediator clarifies the nature of the relationship between the exposure and outcome variables.

Reference: MacKinnon, D. (2012). Introduction to statistical mediation analysis. Routledge.

2) Please mention mechanisms of dairy effects on metabolic syndrome and NAFLD in the abstract. Also add a brief review of previous study.

Response: Thank you very much for your suggestion. Abstract was altered to the following.

It is now read as, “Calcium in dairy products improves the blood lipid profile by increasing HDL cholesterol and reducing triglycerides, while also promoting fat excretion to reduce liver fat accumulation. Additionally, whey protein enhances insulin secretion and improves lipid metabolism in the liver, thereby reducing fat accumulation. Whey protein also reduces postprandial blood glucose levels and suppresses appetite, mitigating key risk factors of metabolic syndrome (MetS). Additionally, probiotics in dairy products improve insulin resistance, while calcium increases HDL cholesterol and helps lower blood pressure. The mechanisms of dairy product consumption affect the MetS components and thereby influence non-alcoholic fatty liver disease (NAFLD) remain unclear. This study aimed to examine the mediating effects of MetS components on dairy product consumption and NAFLD.”

Reference: Soerensen, K.V., et al., Effect of dairy calcium from cheese and milk on fecal fat excretion, blood lipids, and appetite in young men. AJCN, 2014. 99(5): p. 984-991.

Tulipano, G., et al., Whey proteins as source of dipeptidyl dipeptidase IV (dipeptidyl peptidase-4) inhibitors. Peptides, 2011. 32(4): p. 835-838.

Azadbakht, L., et al., Dairy consumption is inversely associated with the prevalence of the metabolic syndrome in Tehranian adults. Am J Clin Nutr, 2005. 82(3): p. 523-30.

Petersen, B.L., et al., A whey protein supplement decreases post-prandial glycemia. Nutr. J., 2009. 8: p. 1-5.

Akhavan, T., et al., Effect of premeal consumption of whey protein and its hydrolysate on food intake and postmeal glycemia and insulin responses in young adults. AJCN, 2010. 91(4): p. 966-975.

van Meijl, L.E., R. Vrolix, and R.P. Mensink, Dairy product consumption and the metabolic syndrome. Nutr. Res. Rev., 2008. 21(2): p. 148-157.

Hajjar, I.M., C.E. Grim, and T.A. Kotchen, Dietary calcium lowers the age‐related rise in blood pressure in the United States: the NHANES III survey. J. Clin. Hypertens., 2003. 5(2): p. 122-126.

Jorde, R. and K.H. Bønaa, Calcium from dairy products, vitamin D intake, and blood pressure: the Tromsø study. Am. J. Clin., 2000. 71(6): p. 1530-1535.

3) It was better you using the 24-h dietary recall third a week (One holiday and two non-holiday days.

Response: The authors greatly appreciate the reviewer’s recommendation.

As mentioned in the Methods and Discussion limitations, the study used the KNHANES dataset, so the authors were not able to change the day of the survey at their discretion. In KNHANES, participants completed the 24-hour dietary recall for only one day.

4) Please add statistical methods under all tables.

Response: The authors greatly appreciate the response. We added statistical methods under all tables (Tables 3-5).

5) Add Antioxidants such as vitamin E, ,Zn, Se, Mg, and amount of different kinds of fatty acids and amino acids to table 2.

Response: Thank you very much for your suggestion.

We conducted additional analyses on antioxidant nutrients, including vitamin A, vitamin E, vitamin C, zinc, and magnesium, as well as fatty acids, including SFA, MUFA, and PUFA. Other nutrients could not be included as they were not assessed in the KNHANES survey.

It now reads “In men, individuals with NAFLD had significantly higher intakes of saturated fatty acids, monounsaturated fatty acids, and total fat compared to the normal group (p = 0.04, p = 0.02, p = 0.02, respectively), whereas polyunsaturated fatty acid intake was significantly lower in the NAFLD group (p = 0.04). Sodium intake was also significantly higher in men with NAFLD (p = 0.03). In women, individuals with NAFLD had significantly lower intakes of saturated fatty acids, monounsaturated fatty acids, polyunsaturated fatty acids, and total fat compared to the normal group (p = 0.008, p = 0.01, p = 0.003, p = 0.003, respectively). Additionally, vitamin E and calcium intake were significantly lower in women with NAFLD (p = 0.03, p = 0.04, respectively).”

6) Please mention others research with more details for comparison their results with Your study results in the discussion.

Response: The authors appreciate the reviewer’s recommendation. We added new contents and references.

“In addition to calcium's role in lipid metabolism, dairy components such as inulin-type fructans and whey protein have been suggested to modulate gut microbiota, potentially contributing to the inhibition of NAFLD progression through the gut-liver axis [21].”

“Consumption of more than 200 g of milk per day was related to a 12% decline in the abdominal obesity risk (95% CI: 0.79–0.97), and consuming more than 100 g of yogurt was related to a 16% reduced hyperglycemia risk (95% CI: 0.70–0.98) [23].”

Particularly, low-fat dairy products intake has been shown to have a positive effect on blood pressure regulation in elderly, and dairy products consumption has also been associated with lower blood pressure in children and adolescents [32, 33]. Furthermore, the consumption of low-fat dairy products has been reported to be more effective in lowering blood pressure, while full-fat dairy products have shown little or no effect [34].

“As indicated by prior research, men have been shown to have a higher tendency for visceral fat accumulation, which is associated with elevated blood pressure. In contrast, women primarily accumulate subcutaneous fat, which is less metabolically active. Consequently, men generally accumulate more visceral fat than women [41-43].”

References:

Reimer RA, Willis HJ, Tunnicliffe JM, Park H, Madsen KL, Soto‐Vaca A. Inulin‐type fructans and whey protein both modulate appetite but only fructans alter gut microbiota in adults with overweight/obesity: a randomized controlled trial. Mol Nutr Food Res. 2017;61(11):1700484.

Lee M, Lee H, Kim J. Dairy food consumption is associated with a lower risk of the metabolic syndrome and its components: a systematic review and meta-analysis. Br J Nutr. 2018;120(4):373-84. Epub 20180606. doi: 10.1017/s0007114518001460. PubMed PMID: 29871703.

Lana A, Banegas JR, Guallar-Castillón P, Rodríguez-Artalejo F, Lopez-Garcia E. Association of dairy consumption and 24-hour blood pressure in older adults with hypertension. Am J Med. 2018;131(10):1238-49.

Yuan WL, Kakinami L, Gray-Donald K, Czernichow S, Lambert M, Paradis G. Influence of dairy product consumption on children's blood pressure: results from the QUALITY cohort. JAND. 2013;113(7):936-41.

Alonso A, Beunza JJ, Delgado-Rodríguez M, Martínez JA, Martínez-González MA. Low-fat dairy consumption and reduced risk of hypertension: the Seguimiento Universidad de Navarra (SUN) cohort. Am J Clin. 2005;82(5):972-9.

Bredella MA. Sex Differences in Body Composition. Adv Exp Med Biol. 2017;1043:9-27.

Van Pelt RE, Jankowski CM, Gozansky WS, Wolfe P, Schwartz RS, Kohrt WM. Sex differences in the association of thigh fat and metabolic risk in older adults. Obesity. 2011;19(2):422-8.

Matsushita Y, Nakagawa T, Yamamoto S, Takahashi Y, Yokoyama T, Noda M, et al. Associations of visceral and subcutaneous fat areas with the prevalence of metabolic risk factor clustering in 6,292 Japanese individuals: the Hitachi Health Study. Diabetes Care. 2010;33(9):2117-9.

Once again, the authors truly appreciate all the constructive comments and suggestions from the reviewer.

Response letter to Reviewer 2

1) Author should discuss their findings rigorously to explain the reason behind the findings.

Response: We appreciate your logical suggestions. The authors should rigorously discuss their findings to explain the biological mechanisms underlying the mediating effects of metabolic syndrome components between dairy product intake and NAFLD, and provide scientific evidence for the gender differences observed in these relationships as written below.

“In addition to calcium's role in lipid metabolism, dairy components such as inulin-type fructans and whey protein have been suggested to modulate gut microbiota, potentially contributing to the inhibition of NAFLD progression through the gut-liver axis [22].”

“Consumption of more than 200 g of milk per day was related to a 12% decline in the abdominal obesity risk (95% CI: 0.79–0.97), and consuming more than 100 g of yogurt was related to a 16% reduced hyperglycemia risk (95% CI: 0.70–0.98) [23].”

Particularly, low-fat dairy products intake has been shown to have a positive effect on blood pressure regulation in elderly, and dairy products consumption has also been associated with lower blood pressure in children and adolescents [32, 33]. Furthermore, the consumption of low-fat dairy products has been reported to be more effective in lowering blood pressure, while full-fat dairy products have shown little or no effect [34].

“As indicated by prior research, men have been shown to have a higher tendency for visceral fat accumulation, which is associated with elevated blood pressure. In contrast, women primarily accumulate subcutaneous fat, which is less metabolically active. Consequently, men generally accumulate more visceral fat than women [41-43].”

References:

Reimer RA, Willis HJ, Tunnicliffe JM, Park H, Madsen KL, Soto‐Vaca A. Inulin‐type fructans and whey protein both modulate appetite but only fructans alter gut microbiota in adults with overweight/obesity: a randomized controlled trial. Mol Nutr Food Res. 2017;61(11):1700484.

Lee M, Lee H, Kim J. Dairy food consumption is associated with a lower risk of the metabolic syndrome and its components: a systematic review and meta-analysis. Br J Nutr. 2018;120(4):373-84. Epub 20180606. doi: 10.1017/s0007114518001460. PubMed PMID: 29871703.

Lana A, Banegas JR, Guallar-Castillón P, Rodríguez-Artalejo F, Lopez-Garcia E. Association of dairy consumption and 24-hour blood pressure in older adults with hypertension. Am J Med. 2018;131(10):1238-49.

Yuan WL, Kakinami L, Gray-Donald K, Czernichow S, Lambert M, Paradis G. Influence of dairy product consumption on children's blood pressure: results from the QUALITY cohort. JAND. 2013;113(7):936-41.

Alonso A, Beunza JJ, Delgado-Rodríguez M, Martínez JA, Martínez-González MA. Low-fat dairy consumption and reduced risk of hypertension: the Seguimiento Universidad de Navarra (SUN) cohort. Am J Clin. 2005;82(5):972-9.

Bredella MA. Sex Differences in Body Composition. Adv Exp Med Biol. 2017;1043:9-27.

Van Pelt RE, Jankowski CM, Gozansky WS, Wolfe P, Schwartz RS, Kohrt WM. Sex differences in the association of thigh fat and metabolic risk in older adults. Obesity. 2011;19(2):422-8.

Matsushita Y, Nakagawa T, Yamamoto S, Takahashi Y, Yokoyama T, Noda M, et al. Associations of visceral and subcutaneous fat areas with the prevalence of metabolic risk factor clustering in 6,292 Japanese individuals: the Hitachi Health Study. Diabetes Care. 2010;33(9):2117-9.

Once again, the authors truly appreciate all the constructive comments and suggestions from the reviewer.

---

## [Decision Letter · Decision Letter 1]

5 Aug 2025

PONE-D-24-31691R1Examining the mediating effects of metabolic syndrome components between dairy products and non-alcoholic fatty liver disease in Korean adultsPLOS ONE

Dear Dr. Shin,

Thank you for submitting your manuscript to PLOS ONE. After careful consideration, we feel that it has merit but does not fully meet PLOS ONE’s publication criteria as it currently stands. Therefore, we invite you to submit a revised version of the manuscript that addresses the points raised during the review process.

We look forward to receiving your revised manuscript.

Kind regards,

Ian James Martins, PhD

Academic Editor

PLOS ONE

Journal Requirements:

Reviewers' comments:

Reviewer's Responses to Questions

**Comments to the Author**

1. If the authors have adequately addressed your comments raised in a previous round of review and you feel that this manuscript is now acceptable for publication, you may indicate that here to bypass the “Comments to the Author” section, enter your conflict of interest statement in the “Confidential to Editor” section, and submit your "Accept" recommendation.

Reviewer #3: All comments have been addressed

2. Is the manuscript technically sound, and do the data support the conclusions?

Reviewer #3: Yes

3. Has the statistical analysis been performed appropriately and rigorously? 

Reviewer #3: No

4. Have the authors made all data underlying the findings in their manuscript fully available?

The PLOS Data policy requires authors to make all data underlying the findings described in their manuscript fully available without restriction, with rare exception (please refer to the Data Availability Statement in the manuscript PDF file). The data should be provided as part of the manuscript or its supporting information, or deposited to a public repository. For example, in addition to summary statistics, the data points behind means, medians and variance measures should be available. If there are restrictions on publicly sharing data—e.g. participant privacy or use of data from a third party—those must be specified.requires authors to make all data underlying the findings described in their manuscript fully available without restriction, with rare exception (please refer to the Data Availability Statement in the manuscript PDF file). The data should be provided as part of the manuscript or its supporting information, or deposited to a public repository. For example, in addition to summary statistics, the data points behind means, medians and variance measures should be available. If there are restrictions on publicly sharing data—e.g. participant privacy or use of data from a third party—those must be specified.requires authors to make all data underlying the findings described in their manuscript fully available without restriction, with rare exception (please refer to the Data Availability Statement in the manuscript PDF file). The data should be provided as part of the manuscript or its supporting information, or deposited to a public repository. For example, in addition to summary statistics, the data points behind means, medians and variance measures should be available. If there are restrictions on publicly sharing data—e.g. participant privacy or use of data from a third party—those must be specified.requires authors to make all data underlying the findings described in their manuscript fully available without restriction, with rare exception (please refer to the Data Availability Statement in the manuscript PDF file). The data should be provided as part of the manuscript or its supporting information, or deposited to a public repository. For example, in addition to summary statistics, the data points behind means, medians and variance measures should be available. If there are restrictions on publicly sharing data—e.g. participant privacy or use of data from a third party—those must be specified.

Reviewer #3: Yes

5. Is the manuscript presented in an intelligible fashion and written in standard English?

Reviewer #3: Yes

6. Review Comments to the Author

Reviewer #3: The authors have addressed most of the previous reviewers' comments thoroughly.The study provides valuable insights into gender-specific mediation pathways between dairy intake, metabolic syndrome (MetS) components, and NAFLD in a large Korean cohort. However, several issues require attention to ensure clarity, statistical robustness, and contextualization of findings.

1. In the Abstract and Results sections (page 13, line 31–32), it is stated that women consuming ≥1 serving of dairy had reduced odds of abdominal obesity (AOR: 0.84, 95% CI: 0.73–0.97). However, in Table 4 (page 23), the AOR for abdominal obesity is listed as 0.84 (95% CI: 0.73–0.97) under the "≥1 serving" group, but the footnote describes this CI as "1.50–2.06" (likely a typographical error).

2. The manuscript states that "all variables used in the mediation analysis were continuous" (page 17, line 147). However, NAFLD (binary outcome) and MetS components (binary/categorical in regression models) were analyzed using logistic regression elsewhere.

3. While covariates (age, BMI, lifestyle factors) were adjusted for in regression models, total energy intake was not included as a covariate in: Mediation analyses (page 17, lines 145–150); Associations between dairy intake and MetS components (Table 4).

4. The study aggregates all dairy types (milk, yogurt, cheese, etc.) but notes differential effects of low-fat vs. full-fat dairy on blood pressure (pages 28–29, 33). Despite this, stratified analyses by dairy type/fat content are limited.

7. PLOS authors have the option to publish the peer review history of their article (what does this mean?). If published, this will include your full peer review and any attached files.). If published, this will include your full peer review and any attached files.). If published, this will include your full peer review and any attached files.). If published, this will include your full peer review and any attached files.

...

Reviewer #3: No

---

## [Author Response · Author response to Decision Letter 2]

12 Aug 2025

Response letter to Reviewer 3

1) In the Abstract and Results sections (page 13, line 31–32), it is stated that women consuming ≥1 serving of dairy had reduced odds of abdominal obesity (AOR: 0.84, 95% CI: 0.73–0.97). However, in Table 4 (page 23), the AOR for abdominal obesity is listed as 0.84 (95% CI: 0.73–0.97) under the "≥1 serving" group, but the footnote describes this CI as "1.50–2.06" (likely a typographical error).

Response: The authors greatly appreciate the reviewer’s careful observation.

We have re-checked Table 4 as well as the Abstract and Results sections and confirmed that the AOR and 95% CI for abdominal obesity in women consuming ≥1 serving of dairy products are consistently reported as 0.84 (95% CI: 0.73–0.97). Therefore, no change was necessary to Table 4.

2) The manuscript states that "all variables used in the mediation analysis were continuous" (page 17, line 147). However, NAFLD (binary outcome) and MetS components (binary/categorical in regression models) were analyzed using logistic regression elsewhere.

Response: Thank you very much for your suggestion.

In the mediation models, the independent variable (dairy product intake) and the mediator variables (MetS components) were entered as continuous measures. NAFLD, although classified as a binary outcome using a cut-off value of >36 from the Hepatic Steatosis Index (HSI) in logistic regression, was represented by its continuous HSI score in the mediation analysis. Using the continuous HSI score allowed us to capture the full spectrum of hepatic steatosis severity, preserve variability, and meet the linearity assumptions of the bootstrapping approach implemented in the SAS macro.

We have added the Methods section to clarify this point as follows: “NAFLD, although classified as a binary outcome using a cut-off value of >36 on the HSI in logistic regression, was represented by its continuous HSI score in the mediation analysis. Using the continuous HSI score allowed us to capture the full spectrum of hepatic steatosis severity, preserve variability, and meet the linearity assumptions of the bootstrapping approach implemented in the SAS macro.”

3) While covariates (age, BMI, lifestyle factors) were adjusted for in regression models, total energy intake was not included as a covariate in: Mediation analyses (page 17, lines 145–150); Associations between dairy intake and MetS components (Table 4).

Response: Thank you for pointing this out.

We would like to clarify that total energy intake was included as a covariate in all regression models, including both the mediation analyses and the analyses of associations between dairy intake and MetS components. The footnotes of Table 4 and Figure 2 already indicate that total energy intake was adjusted for. However, it was inadvertently omitted from the covariate list in the Statistical Analyses section, which may have caused confusion. We have now revised this section to explicitly state “... after controlling for covariates such as age, education level, income, employment status, marital status, household type, region, drinking and smoking status, total energy intake, physical activity, and BMI” to avoid ambiguity.

4) The study aggregates all dairy types (milk, yogurt, cheese, etc.) but notes differential effects of low-fat vs. full-fat dairy on blood pressure (pages 28–29, 33). Despite this, stratified analyses by dairy type/fat content are limited.

Response: We appreciate the reviewer’s comment regarding stratified analyses by dairy fat content.

We acknowledge that low-fat and full-fat dairy products may exert differential physiological effects. However, the KNHANES dataset does not provide detailed fat content information for all dairy subtypes, which limits our ability to perform stratified analyses by fat content. Nevertheless, as noted in the Discussion section, “Some studies have reported that low-fat dairy consumption was inversely associated with blood pressure, whereas the association with full-fat dairy was inconsistent”. While we were unable to perform separate analyses by fat content, by citing these studies, we acknowledge and contextualize potential variations in health effects according to dairy fat content within the interpretation of our findings.

Once again, the authors truly appreciate all the constructive comments and suggestions from the reviewer.

---

## [Decision Letter · Decision Letter 2]

15 Oct 2025

PONE-D-24-31691R2Examining the mediating effects of metabolic syndrome components on the relationship between dairy products and non-alcoholic fatty liver disease in Korean adultsPLOS ONE

Dear Dr. Dayeon Shin,

Thank you for submitting your manuscript to PLOS ONE. After careful consideration, we feel that it has merit but does not fully meet PLOS ONE’s publication criteria as it currently stands. Therefore, we invite you to submit a revised version of the manuscript that addresses the points raised during the review process.

We look forward to receiving your revised manuscript.

Kind regards,

Ian James Martins, PhD

Academic Editor

PLOS ONE

Journal Requirements:

Reviewers' comments:

Reviewer's Responses to Questions

**Comments to the Author**

1. If the authors have adequately addressed your comments raised in a previous round of review and you feel that this manuscript is now acceptable for publication, you may indicate that here to bypass the “Comments to the Author” section, enter your conflict of interest statement in the “Confidential to Editor” section, and submit your "Accept" recommendation.

Reviewer #4: (No Response)

Reviewer #5: All comments have been addressed

2. Is the manuscript technically sound, and do the data support the conclusions?

Reviewer #4: (No Response)

Reviewer #5: Yes

3. Has the statistical analysis been performed appropriately and rigorously? 

Reviewer #4: (No Response)

Reviewer #5: Yes

4. Have the authors made all data underlying the findings in their manuscript fully available?

The PLOS Data policy requires authors to make all data underlying the findings described in their manuscript fully available without restriction, with rare exception (please refer to the Data Availability Statement in the manuscript PDF file). The data should be provided as part of the manuscript or its supporting information, or deposited to a public repository. For example, in addition to summary statistics, the data points behind means, medians and variance measures should be available. If there are restrictions on publicly sharing data—e.g. participant privacy or use of data from a third party—those must be specified.requires authors to make all data underlying the findings described in their manuscript fully available without restriction, with rare exception (please refer to the Data Availability Statement in the manuscript PDF file). The data should be provided as part of the manuscript or its supporting information, or deposited to a public repository. For example, in addition to summary statistics, the data points behind means, medians and variance measures should be available. If there are restrictions on publicly sharing data—e.g. participant privacy or use of data from a third party—those must be specified.requires authors to make all data underlying the findings described in their manuscript fully available without restriction, with rare exception (please refer to the Data Availability Statement in the manuscript PDF file). The data should be provided as part of the manuscript or its supporting information, or deposited to a public repository. For example, in addition to summary statistics, the data points behind means, medians and variance measures should be available. If there are restrictions on publicly sharing data—e.g. participant privacy or use of data from a third party—those must be specified.requires authors to make all data underlying the findings described in their manuscript fully available without restriction, with rare exception (please refer to the Data Availability Statement in the manuscript PDF file). The data should be provided as part of the manuscript or its supporting information, or deposited to a public repository. For example, in addition to summary statistics, the data points behind means, medians and variance measures should be available. If there are restrictions on publicly sharing data—e.g. participant privacy or use of data from a third party—those must be specified.

Reviewer #4: (No Response)

Reviewer #5: Yes

5. Is the manuscript presented in an intelligible fashion and written in standard English?

Reviewer #4: (No Response)

Reviewer #5: Yes

6. Review Comments to the Author

Reviewer #4: (No Response)

Reviewer #5: the new version of this manuscript is OK for publication. So, i do not have additional comments for this version.

7. PLOS authors have the option to publish the peer review history of their article (what does this mean?). If published, this will include your full peer review and any attached files.). If published, this will include your full peer review and any attached files.). If published, this will include your full peer review and any attached files.). If published, this will include your full peer review and any attached files.

...

Reviewer #4: No

Reviewer #5: **Yes:** Mehran RahimlouMehran RahimlouMehran RahimlouMehran Rahimlou

---

## [Author Response · Author response to Decision Letter 3]

28 Oct 2025

Response letter to Reviewer 5

1) In Table3, the analysis of the relationship between dairy product intake and NAFLD relies on a three-category exposure with the lowest category set as the reference. This approach implicitly assumes stepwise monotonic differences between categories and may obscure a non-linear dose–response pattern. It is possible that the association between dairy intake and NAFLD is not linear; therefore, using flexible methods such as restricted cubic splines or fractional polynomials to test for non-linearity in order to reduce the risk of misclassification arising from coarse categorization.

Response: The authors greatly appreciate the reviewer’s careful observation.

As recommended, we conducted an additional restricted cubic spline (RCS) analysis to examine the potential non-linear association between dairy product intake and NAFLD.

The spline model showed a non-linear inverse trend, with a gradual decrease in NAFLD risk at low levels of dairy product intake and a plateau thereafter. However, the overall pattern remained consistent with the results presented in Table 3, and the main findings and conclusions were not materially changed.

Given that the spline analysis did not alter the substantive interpretation, we decided to retain the simple model in the main text for clarity and interpretability, while providing the RCS results and figure in the Supplementary Materials for reference.

This approach allows us to fully address the reviewer’s comment, and the results are visually presented in the Figure 1 below.

Figure 1. Restricted Cubic Spline of Dairy product intake and non-alcoholic fatty liver disease by sex. (Blue indicates men, and red indicates women.)

2) The mediation analysis examining the relationship between dairy product consumption and NAFLD should clarify how medication use was handled, particularly lipid-lowering agents.

Response: Thank you very much for your suggestion.

The Korea National Health and Nutrition Examination Survey (KNHANES 2019–2021) includes a variable indicating whether participants are taking medication for dyslipidemia. However, the dataset does not provide information on which lipid component (e.g., LDL, TG, HDL) the medication specifically targets. Additionally, details such as medication type, dosage, and combination therapy are not collected, making it difficult to accurately distinguish or interpret the effects of such medications. Due to these limitations, this variable was excluded from the mediation model, and the potential for residual confounding has been acknowledged as a limitation in the Discussion section of the manuscript.

It now reads “In addition, although the KNHANES dataset includes a variable indicating whether participants were taking medication for dyslipidemia, it lacks information on medication type, dosage, therapeutic target (e.g., LDL, HDL, TG), and combination therapy, so it could not be considered in the analysis.”.

3) The manuscript should more explicitly address how baseline and concurrent liver status—such as fibrosis stage and liver enzyme levels (e.g., ALT, AST, GGT, ALP)—was handled in the mediation analysis (Figure2), as these factors are closely related to both NAFLD development/progression and potential exposures/mediators. Without clear treatment of these variables, there is a risk of confounding

Response: Thank you for pointing this out.

In this study, non-alcoholic fatty liver disease (NAFLD) was defined using the Hepatic Steatosis Index (HSI), which incorporates the ALT/AST ratio, body mass index (BMI), sex, and diabetes status. Because ALT and AST are integral components of the HSI formula, they were not included as separate covariates in the mediation analysis.

Unfortunately, the Korea National Health and Nutrition Examination Survey (KNHANES 2019–2021) dataset used in this study does not contain data on GGT, ALP, or fibrosis-related indices such as FIB-4 or NFS. Therefore, these variables could not be included in our analyses.

It now reads “Furthermore, the dataset does not provide information on liver fibrosis stage or liver enzyme levels such as GGT or ALP, which may also influence NAFLD risk.”.

Once again, the authors truly appreciate all the constructive comments and suggestions from the reviewer.

---

## [Decision Letter · Decision Letter 3]

3 Dec 2025

PONE-D-24-31691R3Examining the mediating effects of metabolic syndrome components on the relationship between dairy products and non-alcoholic fatty liver disease in Korean adultsPLOS ONE

Dear Dr. Shin,

Thank you for submitting your manuscript to PLOS ONE. After careful consideration, we feel that it has merit but does not fully meet PLOS ONE’s publication criteria as it currently stands. Therefore, we invite you to submit a revised version of the manuscript that addresses the points raised during the review process.

We look forward to receiving your revised manuscript.

Kind regards,

Ian James Martins, PhD

Academic Editor

PLOS ONE

Journal Requirements:

Additional Editor Comments:

PLEASE RESPOND TO REVEIWER 6'S COMMENTS

Reviewers' comments:

Reviewer's Responses to Questions

**Comments to the Author**

1. If the authors have adequately addressed your comments raised in a previous round of review and you feel that this manuscript is now acceptable for publication, you may indicate that here to bypass the “Comments to the Author” section, enter your conflict of interest statement in the “Confidential to Editor” section, and submit your "Accept" recommendation.

Reviewer #4: All comments have been addressed

Reviewer #6: All comments have been addressed

2. Is the manuscript technically sound, and do the data support the conclusions?

Reviewer #4: Yes

Reviewer #6: Yes

3. Has the statistical analysis been performed appropriately and rigorously? 

Reviewer #4: Yes

Reviewer #6: Yes

4. Have the authors made all data underlying the findings in their manuscript fully available?

The PLOS Data policy requires authors to make all data underlying the findings described in their manuscript fully available without restriction, with rare exception (please refer to the Data Availability Statement in the manuscript PDF file). The data should be provided as part of the manuscript or its supporting information, or deposited to a public repository. For example, in addition to summary statistics, the data points behind means, medians and variance measures should be available. If there are restrictions on publicly sharing data—e.g. participant privacy or use of data from a third party—those must be specified.requires authors to make all data underlying the findings described in their manuscript fully available without restriction, with rare exception (please refer to the Data Availability Statement in the manuscript PDF file). The data should be provided as part of the manuscript or its supporting information, or deposited to a public repository. For example, in addition to summary statistics, the data points behind means, medians and variance measures should be available. If there are restrictions on publicly sharing data—e.g. participant privacy or use of data from a third party—those must be specified.requires authors to make all data underlying the findings described in their manuscript fully available without restriction, with rare exception (please refer to the Data Availability Statement in the manuscript PDF file). The data should be provided as part of the manuscript or its supporting information, or deposited to a public repository. For example, in addition to summary statistics, the data points behind means, medians and variance measures should be available. If there are restrictions on publicly sharing data—e.g. participant privacy or use of data from a third party—those must be specified.requires authors to make all data underlying the findings described in their manuscript fully available without restriction, with rare exception (please refer to the Data Availability Statement in the manuscript PDF file). The data should be provided as part of the manuscript or its supporting information, or deposited to a public repository. For example, in addition to summary statistics, the data points behind means, medians and variance measures should be available. If there are restrictions on publicly sharing data—e.g. participant privacy or use of data from a third party—those must be specified.

Reviewer #4: Yes

Reviewer #6: Yes

5. Is the manuscript presented in an intelligible fashion and written in standard English?

Reviewer #4: Yes

Reviewer #6: Yes

6. Review Comments to the Author

Reviewer #4: (No Response)

Reviewer #6: The manuscript explores how metabolic syndrome components may explain the link between dairy product intake and non-alcoholic fatty liver disease (NAFLD) in Korean adults, using a large dataset. The title, abstract, and introduction are clear and appropriate. Methods and statistical analyses are well described, though some limitations such as reliance on self-reported dietary data and indirect definition of NAFLD are noted. Results are presented clearly with helpful tables, and sex-specific effects are addressed. The discussion fairly interprets the findings within the context of existing studies and recognizes the study’s main limitations, including its cross-sectional design. Authors have been revised manuscript based on the previous commnets.Some minor issues still seem unclear. Below are several constructive comments for your consideration.

1-Please consider adding sensitivity analyses that use alternative NAFLD indices where possible, or analyses stratified by key variables like age and BMI.

2-It would strengthen the paper to discuss the possible biological reasons for the sex differences observed—such as hormonal or genetic factors involved in metabolic syndrome.

3-Also, please expand on how the serving sizes used in your study relate to clinical practice, and discuss whether your findings can be applied to populations outside Korea, given the differences in dairy products across countries.

7. PLOS authors have the option to publish the peer review history of their article (what does this mean?). If published, this will include your full peer review and any attached files.). If published, this will include your full peer review and any attached files.). If published, this will include your full peer review and any attached files.). If published, this will include your full peer review and any attached files.

...

Reviewer #4: No

Reviewer #6: No

---

## [Author Response · Author response to Decision Letter 4]

23 Jan 2026

Title: Examining the mediating effects of metabolic syndrome components on the relationship between dairy product consumption and nonalcoholic fatty liver disease in Korean adults

Response letter to Reviewer 6

We highly appreciate your insightful and helpful comments. Through your comments, we could improve the overall quality of our manuscript. However, prior to the final submission, we carefully reanalyzed the data to ensure reproducibility and identified several minor errors, which have now been corrected. Accordingly, we thoroughly reviewed the manuscript to reflect these changes and to incorporate the reviewers’ suggestions as fully as possible. All page and line numbers refer to the revised manuscript file without track changes.

Comment 1) Please consider adding sensitivity analyses that use alternative NAFLD indices where possible, or analyses stratified by key variables like age and BMI.

Response: We appreciate the reviewer’s constructive suggestion.

In response, we conducted sensitivity analyses stratified by age and BMI to evaluate the robustness of the mediation effects. The results were added as Supplementary Tables.

Across age and BMI strata, the overall pattern of mediation remained consistent with the main findings, and the number of statistically significant natural indirect effects (NIEs) was consistently greater among women than among men, supporting the robustness of the observed sex-specific associations.

“Sensitivity analyses using age- and BMI-stratified mediation models were conducted, with a greater number of NIEs observed among women than among men across strata (Supplementary tables 1 and 2).”

Comment 2) It would strengthen the paper to discuss the possible biological reasons for the sex differences observed—such as hormonal or genetic factors involved in metabolic syndrome.

Response: We thank the reviewer for this helpful comment.

In response, we have expanded the Discussion to address the potential biological mechanisms underlying the observed sex differences. Specifically, we added a new paragraph describing sex-specific metabolic responses to dairy consumption, including differences in adipose tissue distribution, insulin sensitivity, and lipid metabolism. We also incorporated evidence from clinical trials demonstrating differential metabolic responses to dairy intake between men and women.

“Sex-specific associations of dairy consumption with metabolic syndrome and NAFLD”

“The present study demonstrated that the associations between dairy consumption, NAFLD, and MetS components vary by sex, with significant associations observed only among women. Similar sex-specific patterns have been reported in previous studies. A cohort study found that higher dairy consumption was more strongly associated with a lower prevalence of NAFLD in women [45], a finding that was supported by a meta-analysis of observational studies reporting consistent results [46]. Regarding MetS components, a 10-year cohort study from the Korean Genome and Epidemiology Study (KoGES) showed that dairy consumption was associated with a reduced risk of low HDL-cholesterol among women [47], and a cross-sectional study using KNHANES observed a lower MetS prevalence among Korean women consuming ≥1 serving/day of dairy products [48]. Moreover, MetS and its individual components have been shown to be more strongly associated with NAFLD in women than in men [49].

These sex differences may partly arise because the metabolic effects of diets and dietary constituents are expressed in a sex-specific manner and are highly dependent on disease context [50]. Clinical evidence further supports sex-specific metabolic responses to dairy consumption. In a 6-week randomized controlled trial (RCT), low-fat dairy intake differentially improved MetS parameters by sex, with a modest but significant reduction in fasting blood glucose in men and reductions in waist circumference, body weight, and BMI in women [51]. Another RCT demonstrated distinct postprandial lipoprotein responses to the dairy between sexes, with women exhibiting smaller increases in VLDL particles and greater increases in HDL particle concentrations than men [52]. Biological sex also exerts a strong influence on adipose tissue distribution and metabolic function. Women generally have greater amounts of subcutaneous adipose tissue than men [53], and although abdominal subcutaneous fat has been associated with adverse glucose and lipid profiles [54], subcutaneous fat may provide partial protection against metabolic dysfunction, thereby potentially reducing the risk of MetS in women [55]. In addition, insulin sensitivity differs by sex, with women typically exhibiting greater adipose tissue insulin sensitivity than men [56]. Consistently, dairy intake has been shown to inhibit intestinal fat absorption, improve lipid profiles, reduce blood pressure, and enhance glucose metabolism [57-59], thereby contributing to improvements in NAFLD. Sex hormones further modulate these processes, with estrogens promoting metabolically protective pathways and androgens supporting male-specific metabolic environments [60]. Collectively, these biological and clinical differences suggest that dairy exposure may engage distinct metabolic pathways in men and women, contributing to the observed sex-specific associations.”

Comment 3) Also, please expand on how the serving sizes used in your study relate to clinical practice, and discuss whether your findings can be applied to populations outside Korea, given the differences in dairy products across countries.

Response: We are grateful for the reviewer’s thoughtful recommendation.

We have expanded the Discussion to address the clinical relevance of our serving size criteria and the generalizability of our findings to populations outside Korea.

The serving sizes used in this study (200 g for milk, 100–150 g for yogurt, and 20 g for cheese) are consistent with standard portions recommended in international dietary guidelines [26, 61, 62] and commonly used in clinical nutrition practice. These amounts represent realistic, achievable targets for dietary interventions. For instance, 200 g of milk is equivalent to a standard glass, making our findings directly translatable to practical dietary recommendations. Regarding generalizability, while we acknowledge that specific dairy product preferences vary across countries, our analytical approach enhances the external validity of our findings. Our primary mediation analyses focused on total dairy intake, which captures the overall contribution of dairy consumption regardless of specific products consumed. We also conducted sensitivity analyses examining individual dairy types (milk, yogurt, and cheese separately), allowing interpretation across diverse dietary contexts. However, we acknowledge that the relatively low cheese consumption in our Korean cohort may limit the precision of cheese-specific estimates when applied to populations with higher cheese intake patterns.

“The serving sizes used in this study (200 g for milk, 100–150 g for yogurt, and 20 g for cheese) are consistent with standard portions in international dietary guidelines [26, 61, 62] and represent realistic targets for clinical dietary counseling. Although dairy product preferences differ across countries, our focus on total dairy intake in the primary mediation analysis supports the broader applicability of our findings. Sensitivity analyses examining individual dairy types allow for interpretation across diverse dietary contexts, though the generalizability y of cheese-specific findings may be limited given the relatively low cheese consumption in our Korean cohort compared to Western populations.”

Once again, the authors truly appreciate all the constructive comments and suggestions from the reviewer.

---

## [Editor Report · Decision Letter 4]

24 Mar 2026

Examining the mediating effects of metabolic syndrome components on the relationship between dairy product consumption and nonalcoholic fatty liver disease in Korean adults

PONE-D-24-31691R4

Dear Dr. Dayeon Shin,

We’re pleased to inform you that your manuscript has been judged scientifically suitable for publication and will be formally accepted for publication once it meets all outstanding technical requirements.

Kind regards,

Ian James Martins, PhD

Academic Editor

PLOS One
---

## [Editor Report · Acceptance letter]

PONE-D-24-31691R4

PLOS One

Dear Dr. Shin,

I'm pleased to inform you that your manuscript has been deemed suitable for publication in PLOS One. Congratulations! Your manuscript is now being handed over to our production team.

Kind regards,

on behalf of

Dr. Ian James Martins

Academic Editor

PLOS One